# Discovery of synthetic lethal interactions from large-scale pan-cancer perturbation screens

Sumana Srivatsa [1,2,7], Hesam Montazeri [3,7], Gaia Bianco[4,7], Mairene Coto-Llerena[4,5], Mattia Marinucci [4], Charlotte K. Y. Ng [2,6], Salvatore Piscuoglio [4,5] ✉ & Niko Beerenwinkel [1,2] ✉

The development of cancer therapies is limited by the availability of suitable drug targets. Potential candidate drug targets can be identified based on the concept of synthetic lethality (SL), which refers to pairs of genes for which an aberration in either gene alone is non-lethal, but co-occurrence of the aberrations is lethal to the cell. Here, we present SLIdR (Synthetic Lethal Identification in R), a statistical framework for identifying SL pairs from large-scale perturbation screens. SLIdR successfully predicts SL pairs even with small sample sizes while minimizing the number of false positive targets. We apply SLIdR to Project DRIVE data and find both established and potential pan-cancer and cancer type-specific SL pairs consistent with findings from literature and drug response screening data. We experimentally validate two predicted SL interactions (*ARID1A-TEAD1* and *AXIN1-URI1*) in hepatocellular carcinoma, thus corroborating the ability of SLIdR to identify potential drug targets.

Synthetic lethality (SL) refers to gene pairs for which an aberration in either gene alone does not affect cell viability, but aberrations in both genes are fatal to the cell. Key to exploiting SL in cancer therapy is the identification of a targetable dependent gene (SL partner) for a given genetically altered gene, such that the two genes form an SL pair (Fig. 1a). A classical example of SL in cancer therapy is the use of *PARP* inhibitors in *BRCA*-mutated cancers. The *BRCA1/2* genes involved in DNA double-strand break repair are often mutated in breast and ovarian cancers[1–3], and hence such cancer cells rely on alternate DNA repair processes. *PARP1* plays a central role in these alternate DNA repair mechanisms[4,5], and therefore inhibiting *PARP* results in catastrophic double-strand breaks during replication, ultimately leading to cancer cell death[6,7].

In recent years, high-throughput experiments have enabled the generation of multi-omics observational data and large-scale interventional data in various cancer types. Several computational methods have since been developed to identify and prioritize SL interactions from such diverse molecular data. Early methods involved the identification of mutually exclusive genetic alterations in functionally related genes. These approaches were either de novo and detected patterns in the mutational data[8–11], or knowledge-driven and based on pathways or interaction networks[12]. Other computational methods for identifying SL interactions relied on human orthologues of yeast genetic interactions[13], inter-species network models[14], signaling networks[15], and protein-protein interaction networks[16].

Experimentally, large-scale perturbation screens based on siRNA, shRNA, CRISPR, or small molecules in cell lines have been conventionally used to identify SL interactions[17–27]. McDonald et al.[27] conducted a large-scale deep RNAi screen, entitled Project DRIVE, that targeted 7,837 genes in 398 Cancer Cell Line Encyclopedia[28] (CCLE)

[1]Department of Biosystems Science and Engineering, ETH Zurich, 4058 Basel, Switzerland. [2]SIB Swiss Institute of Bioinformatics, Lausanne, Switzerland. [3]Department of Bioinformatics, Institute of Biochemistry and Biophysics, University of Tehran, Tehran, Iran. [4]Visceral Surgery and Precision Medicine Research Laboratory, Department of Biomedicine, University of Basel, 4031 Basel, Switzerland. [5]Institute of Medical Genetics and Pathology, University Hospital Basel, 4031 Basel, Switzerland. [6]Department for BioMedical Research, University of Bern, 3008 Bern, Switzerland. [7]These authors contributed equally: Sumana Srivatsa, Hesam Montazeri, Gaia Bianco. ✉e-mail: s.piscuoglio@unibas.ch; niko.beerenwinkel@bsse.ethz.ch

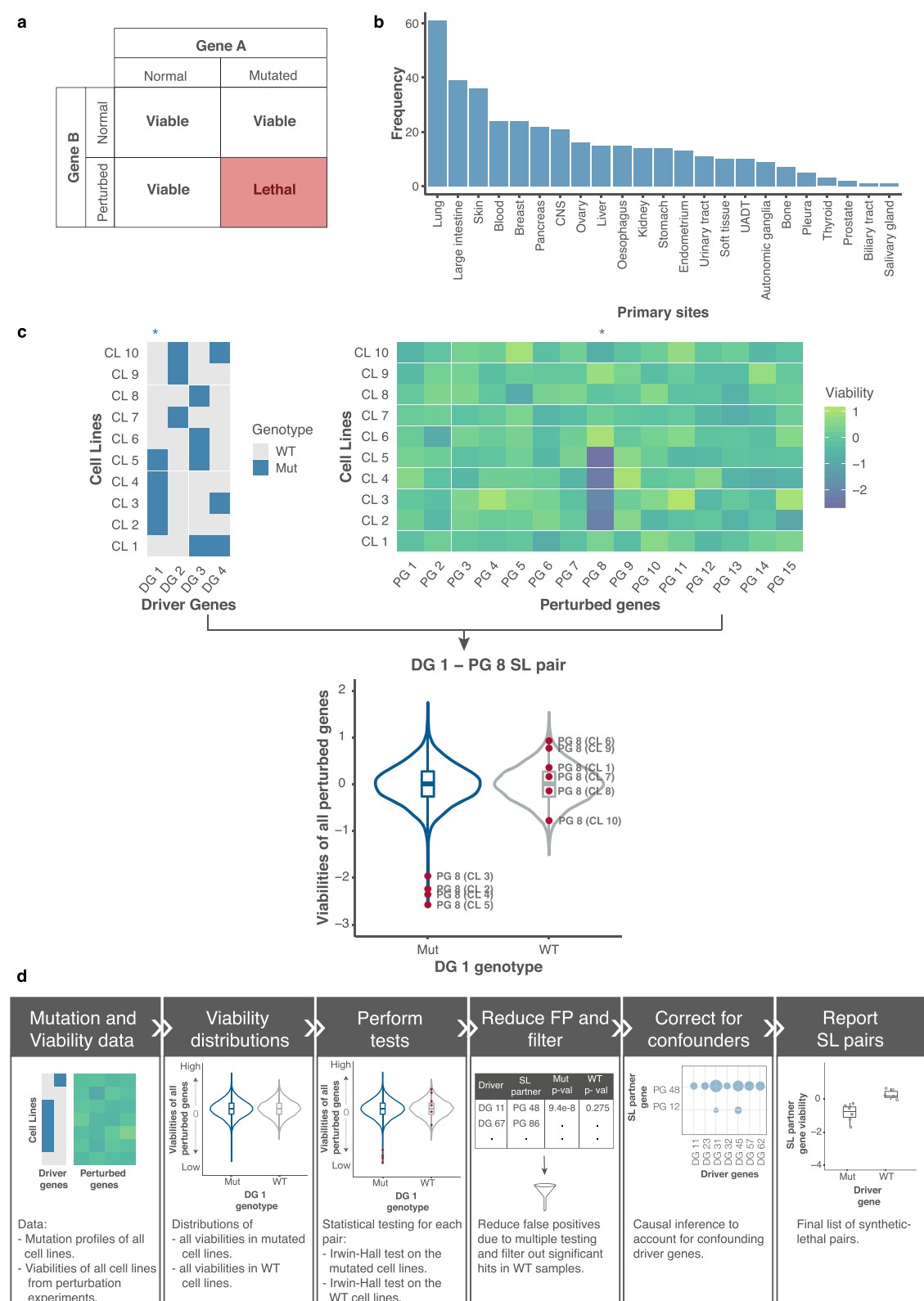

**Fig. 1 | Overview and SLIdR workflow. a** Definition of a synthetic lethal pair: Aberration of gene A (driver gene) or knockdown of gene B (SL partner gene) alone does not affect the viability of the cell. However, the combination of mutated gene A and knockdown of gene B is lethal to the cell. **b** Distribution of the number of cell lines with copy number data from CCLE across 23 different cancer types used in this study. Since most cancers have small sample sizes, the figure indicates the necessity of developing a computational tool with sufficient statistical power on small sample size data. **c** Illustration of the SLIdR algorithm with a toy example. The data consists of driver genes DG 1-DG 4 and perturbed genes PG 1-PG 15 across cell lines CL 1-CL 10. Cell lines CL 2-5 are mutated in the driver gene DG 1 (Mut), while the remaining cell lines are DG 1 wild-type (WT). Comparison of viability distributions across all perturbed genes PG 1-PG 15 in the DG 1 mutated (Mut) and WT cell lines shows that perturbation of gene 8 (PG 8) results in reduced viability only in CL 2-5 and not the WT cell lines. Thus, PG 8 is a SL partner of DG 1. **d** The computational pipeline illustrating the different steps performed to obtain the candidate SL pairs from mutation profiles and perturbation screen data.

models and provided a rich and robust dataset for the identification of SL pairs. However, the authors primarily analyzed gene interactions in a pan-cancer manner.

Significant efforts have been made in developing integrative computational methods that harness both multi-omics observational data and genetic perturbation screening data. Jerby-Arnon et al.[29] proposed DAISY, a method that predicts a global network of potential SL interactions in human cells from tumor copy number, expression data, and shRNA screening data from cell lines. Similarly, Sinha et al.[30] presented MiSL, an algorithm based on Boolean implications to prioritize mutation-specific SL partners for different cancer types from large pan-cancer multi-omics and perturbation screening data. ISLE is another statistical approach that uses lab-screened SL interactions as inputs and analyzes tumor molecular profiles, patient clinical data, and gene phylogeny relations to identify clinically relevant SL interactions[31].

While all these methods mark a significant step forward, general prediction of SL interactions in cancer still remains a challenge. Most methods primarily rely on large sample sizes and multi-omics data, and thus are not suitable for rare cancer types. Furthermore, they use genetic perturbation screening data merely to refine the candidate list of SL pairs derived from analyzing multi-omics data. We hypothesize that such rich large-scale perturbation screens can be exploited further to obtain SL pairs.

Here, we present a statistical framework called SLIdR (Synthetic Lethal Identification in R), a rank-based statistical method for predicting SL pairs from perturbation screens in both pan-cancer and cancer type-specific settings. The predicted SL pairs are validated by large-scale drug-response profiles and literature evidence. Subsequently, by benchmarking SLIdR on simulated data and experimentally identified SL interactions, we demonstrate the improved predictive power and advantage of SLIdR in reducing the number of false positives even when the sample size is small. Finally, we validate two SLIdR predictions in hepatocellular carcinoma, namely *ARID1A-TEAD1* and *AXIN1-URI1*, through comprehensive experiments in patient-derived cell lines.

## Results

### SLIdR workflow

We developed SLIdR, a statistical framework for identifying SL interactions between a genetically altered gene and a perturbed gene from large-scale perturbation screens of cancer cell lines. We focused on significantly mutated genes reported by MutSig 2CV v3.1[32,33] for each cancer type and considered these genes to be genetically altered in cell lines if they were subject to non-synonymous mutations or deep deletions (see Methods). We collectively refer to these altered genes as driver genes and their alterations as mutations. SLIdR aims to find SL partners for such drivers from perturbation data. We applied SLIdR to the Project DRIVE dataset[27], focusing on cell lines from CCLE[28] across various cancer types with available copy number data (Fig. 1b).

In contrast to previous methods which perform statistical tests on the raw viability readouts[27,29,31], SLIdR uses the normalized ranks of the viabilities across all perturbed genes, for each cell line, in order to increase statistical power for small sample sizes. For each driver gene, SLIdR first stratifies the cell lines into mutated and wild-type based on the mutation status of the driver gene. Subsequently, it tests, for each perturbed gene, whether the perturbation results in lower ranked viabilities in the mutated cell lines but not in the wild-type cell lines (Fig. 1c). SLIdR uses two Irwin-Hall tests to mine for such driver-perturbed SL gene pairs (see Methods). Cell lines with several co-occurring driver mutations can yield multiple SL pairs with the same perturbed gene. To identify the most likely SL pairs, we perform causal inference using matching-based potential outcome models. For a given candidate pair, we match the wild-type to mutated cell lines based on the other co-occurring mutations, thus achieving a covariate

balance. Finally, SLIdR compares the viabilities of the matched groups, and the significant SL pairs are reported (Fig. 1d; see Methods).

### Enrichment of pan-cancer SL interactions by SLIdR

To identify pan-cancer SL interactions, we first applied SLIdR to the DRIVE data in a pan-cancer setting. We identified 151 SL pairs (Supplementary Data 1) involving 84 driver genes (Fig. 2a). Out of the 151 SL pairs, five pairs involving bona fide driver genes *TP53*, *KRAS*, *BRAF*, *CTNNB1*, and *PIK3CA* exhibited oncogene-addiction, i.e., they paired with themselves as the SL partner gene. This proved to be an efficient quality check for our method as these are well-established drivers and their subsequent knockdown resulted in cellular mortality. We also found that some cell lines with several co-occurring mutations resulted in multiple driver genes pairing with the same SL partner (Fig. 2b; Supplementary Fig. 1a). For example, co-deletion of genes near p16 including *MTAP* and several interferons is common in several cancers, and subsequently all these drivers paired with *MAT2A* as the SL partner. Using causal inference (see Methods), we accounted for these co-occurring mutations and predicted the relevant driver genes for each SL partner (Fig. 2c; Supplementary Fig. 1b), resulting in 90 SL pairs across 42 driver genes (Fig. 2d).

Top predictions of SLIdR included *PRMT5*, *MAT2A*, and *RIOK1* as SL partners of *MTAP* which are all well-established vulnerable targets for *MTAP*-altered cells[27,34]. SLIdR also predicted *E2F3* and *SKP2* as SL partners of *RB1*[27]. Furthermore, *RPL22* showed lethality with its paralog *RPL22L1* confirming the findings of McDonald et al.[27]. *PIK3CA-BIRC5* was another reassuring pair as depletion of survivin (*BIRC5*) has been shown to have a pro-apoptotic effect in breast cancer cells with *PIK3CA* mutations[35,36]. In addition to established pairs, SLIdR also predicted several potential SL pairs, such as *KRAS-TRPM7* and *TP53*-specific SL partners, including *TP53BP1*, *USP28*, *DDX3*, and *PNPLA6*, which were further supported by evidence in the literature (Supplementary Data 2).

Next, we systematically validated the predicted pan-cancer SL pairs on the primary PRISM repurposing dataset, a large-scale drug-response profile from the DepMap consortium[37]. Out of the predicted SL interactions, 25 SL pairs had at least one matching drug compound in the screen. Stratifying the cell lines based on the mutation status of the driver gene for each pair, we tested if the drug targeting the corresponding SL partner gene was more effective in the mutated cell lines using a t-test (Supplementary Data 3). Reassuringly, we found 13 SL pairs, including *BRAF-CYP2B6*, *BRAF-HTR2C*, *KEAP1-NFE2L2*, *PIK3CA-BIRC5*, and *TP53-GABRR3*, with significant difference in drug response (significance level $\alpha = 0.1$). Of these 13 pairs, *BRAF-HTR2C*, *KEAP1-ATP1A1*, *NOTCH3-WEE1*, and *POTEF-PLK1* were significant after multiple testing correction (q-values ≤ 0.2) across 9 different candidate drug compounds (Supplementary Fig. 2, Supplementary Data 3). The lower coverage after FDR correction can be attributed to the heterogeneity and noise of these screens. As a control experiment, we performed permutation tests across 1000 sets of random gene pairs and found these validated hits to be significant (empirical p-value ≤ 0.005), thereby confirming the predictions from SLIdR (see Methods).

Pan-cancer analyses offer large sample sizes and the ability to identify shared targets across different cancer types. While the latter property is preferred especially from a therapeutic perspective, it is often difficult to identify such pairs due to the inherent genetic diversity in tumors based on their primary sites. To assess the differential sensitivities of the predicted pan-cancer hits based on primary sites, we computed the SLIdR p-values for these predicted pan-cancer SL pairs in subsets of cell lines grouped by primary sites. We found that a sizable fraction of the pan-cancer signals is indeed cancer type-specific (Fig. 2d). For example, SLIdR identified *NFE2L2* as the SL partner of the mutated *KEAP1*, both of which play an important role in cancer through Nrf2 pathway activation[38]. However, this SL interaction was largely driven by lung cancer samples (Fig. 2d), in accordance with

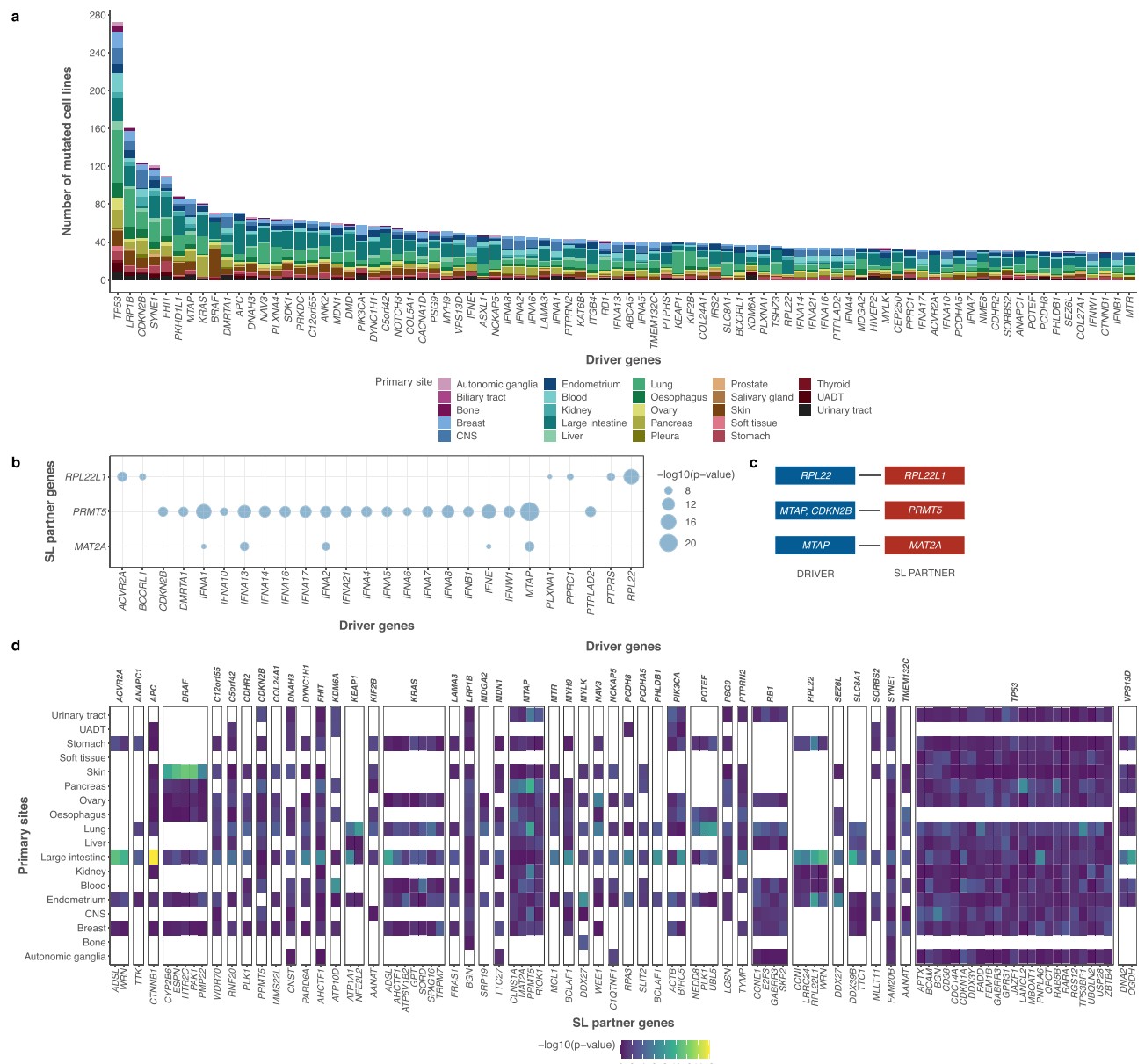

**Fig. 2 | Pan-cancer SLIdR predictions. a** Stacked barplot indicating the frequencies of 84 mutated driver genes across different cancer types. The number of mutated cell lines of a given cancer type may have an impact on the statistical power of the SLIdR framework. **b** Bubble-plot summarizing the significance (-log10(*p*-value)) of different driver genes (x-axis) pairing with the same SL partner gene (y-axis) as predicted by SLIdR in the pan-cancer analysis after filtering out false positives from multiple testing. The *p*-values are computed using one-sided IH-test.
**c** Corresponding list of significant SL pairs after accounting for confounding

mutations and performing causal inference using matching-based potential outcome models. **d** Differential sensitivities of pan-cancer SL pairs in subsets of cell lines grouped by primary sites (y-axis). Each panel corresponds to a specific driver gene (x-axis top) and encapsulates the significance profiles of all its SL-partners (x-axis bottom) across various primary sites. Each column in a given panel depicts the significance profile of the SL pair in subsets of cell lines grouped by primary sites. The *p*-values are computed using one-sided IH-test.

Leiserson et al.[39] who also reported the pair to be mutually exclusive in their pan-cancer TCGA analysis largely due to lung cancer samples. Similarly, signals for *APC-CTNNB1* and all the *BRAF* associated SL interactions were mostly specific to large intestine and skin samples, respectively (Fig. 2d). Thus, although a considerable number of pan-cancer hits are consistent with previous findings, these examples show the need to identify cancer type-specific SL partners.

**Enrichment of cancer type-specific SL interactions by SLIdR**
To identify cancer type-specific SL interactions, we applied SLIdR to the DRIVE data for 17 cancer types and identified a total of 839 SL pairs over 233 unique driver genes (Supplementary Data 1). Out of the 233

drivers, 66 genes were mutated in more than one cancer type (Fig. 3a). However, the mutation profiles are diverse across cancer types, with *TP53* mutations being highly prevalent and observed in 81% of the cancer types, while well-known drivers such as *BRAF*, *APC*, and *PTEN* were distinctly associated with skin, large intestine, and endometrial cancers, respectively.

Upon extensive literature survey of the SL pairs, we identified 55 established and potential pairs with literature support (Fig. 3b; Supplementary Data 2). For example, SLIdR predicted *GATA3-ESR1* in breast cancer. *GATA3* is mutated in >10% of breast cancers and directly impacts *ESR1* enhancer accessibility, thereby altering binding potential and transcriptional targets in tumor cells[40]. Furthermore, *GATA3*

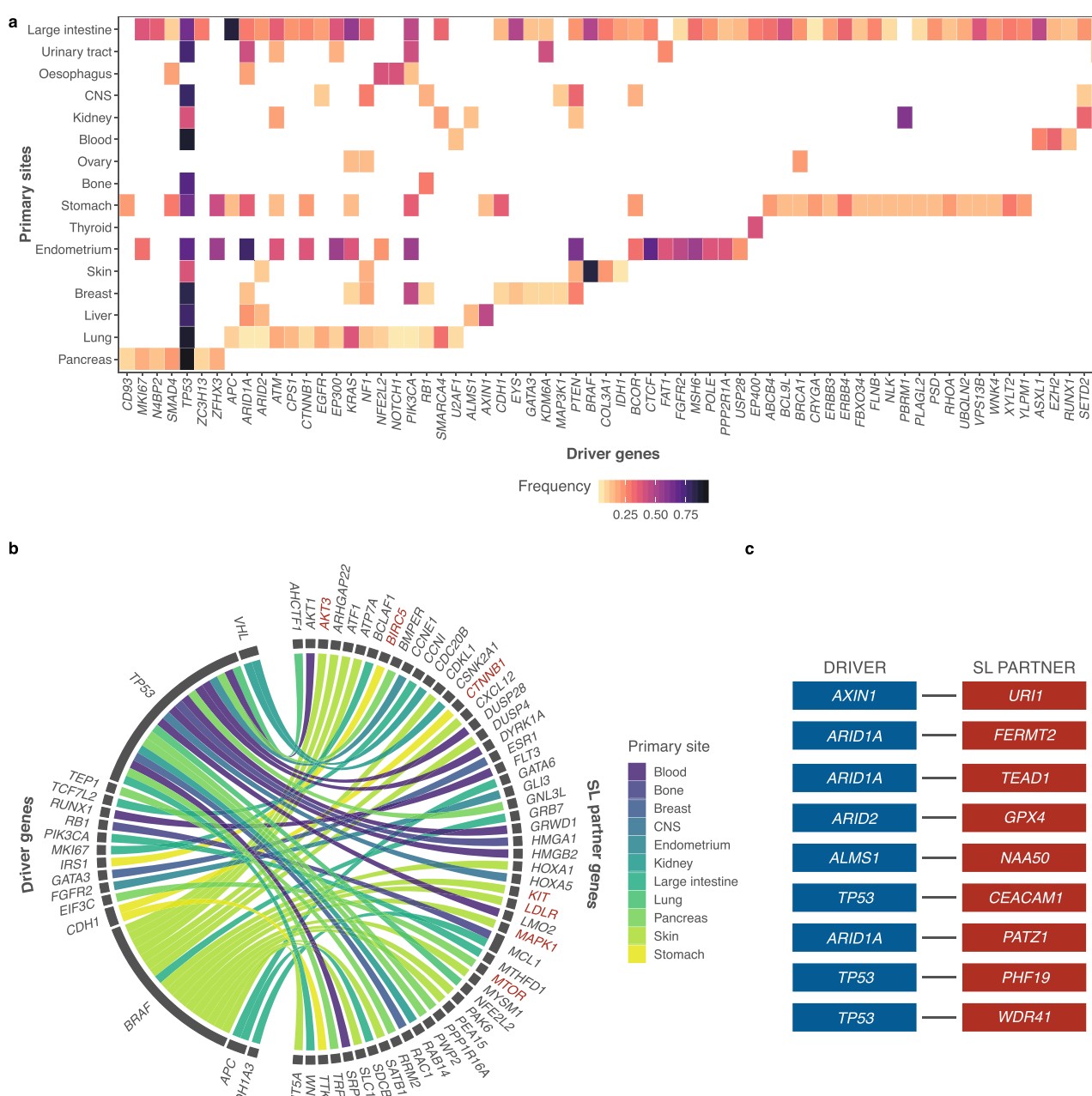

**Fig. 3 | Cancer type-specific SLIdR predictions. a** Heatmap of frequencies of 66 driver genes across 16 cancer types. **b** Circos plot summarizing the SL partners (right) of different driver genes (left) with literature evidence, across 11 cancer types. The SL pairs constituting SL partners in red have further evidence in the PRISM drug response screen (significance level $\alpha = 0.1$). **c** Top-ranked SL pairs in hepatocellular carcinoma reported by SLIdR.

mutations are almost never observed in ER-negative breast cancers, strongly suggesting synthetic lethality. Despite the small sample size (7 cell lines, Fig. 1b), SLIdR also successfully elicited the *RB1-MCL1* pair in osteosarcomas. Loss-of-function *RB1* mutations are common in osteosarcomas[41,42] and inhibition of *MCL1* has been shown to block tumor growth in osteosarcoma[43]. Additionally, SLIdR predicted several significant SL partners specific to *TP53*, including *HMGA1, RAB14, and RAC1* in osteosarcoma, renal, and breast cancers, respectively (Fig. 3b; Supplementary Data 2).

Validating the cancer type-specific predictions on the PRISM drug-response dataset further bolstered our findings and provided druggable targets for 10 cancer types (Supplementary Fig. 2, Supplementary Data 3). Analogous to the pan-cancer analysis for validating predicted SL pairs, we used *t*-test to test the differential drug response between the mutated and WT cell lines. We identified one significant

candidate drug compound each in central nervous system, skin, stomach, and thyroid cancers, and six and eight significant candidate drug compounds in large intestine and pancreatic cancers, respectively (q-values ≤ 0.2). In particular, four of the eight candidate compounds in pancreatic cancer were MTOR-targeting drugs and were significantly more effective in *EIF3C* mutated cell lines than WT cell lines (Supplementary Data 3). It has been previously reported that EIF3 complex instability is linked to deregulation of *MTOR*, leading to increased translation of oncogenic proteins and malignant transformation in pancreatic cancer[44,45], further establishing this interaction in pancreatic cancer. In skin cancer, in addition to one significant hit after FDR correction, BRAF-specific SL partners including *AKT3, KIT, LDLR*, and *MAPK1* found support in the PRISM dataset (significance level $\alpha = 0.1$) and in the literature (Supplementary Data 2, 3). We also performed control experiments based on permutation tests of 1000 sets of

random gene pairs and found some evidence for these hits in pancreatic, skin, stomach, and renal cancers (empirical p-value = {0.024, 0.076, 0.002, 0.075}, respectively), which reached statistical significance after multiple testing correction for stomach cancer (q-value ≤ 0.02). These results demonstrate that SLIdR is capable of finding established and potential cancer type-specific SL pairs.

## SLIdR outperforms conventional tests for SL prediction

To demonstrate the advantage of relative ranking over raw viability scores and to benchmark the performance of SLIdR, we compared SLIdR with existing SL prediction methods in a simulation study. DAISY and ISLE are two popular methods that both use the Wilcoxon rank sum test for predicting candidate SL interactions from perturbation screens[29,31]. While both of these methods are multi-step and use other data sources to predict the final pairs, their reliance on raw viability scores rather than relative viabilities from shRNA screens limits the full utilization of such screens. In fact, the authors of DAISY[29] expressed that the data obtained from shRNA screens have low statistical power and hence, DAISY uses such screens only to refine a list of highly statistically significant SL interactions obtained from other data sources. By contrast, we show here that these screens are more powerful if ranked viabilities across genes are used rather than raw viabilities. In addition, we included the t-test in our simulation study as we found it to be statistically more powerful than the Wilcoxon rank sum test. The simulation study focused on liver, ovarian, and bone cancers. For each cancer type, the ground truth comprised 30 random pairwise interactions between driver genes and perturbed genes. For the simulation study, we reused the binarized mutation matrices of the corresponding cancer types and simulated the viabilities by sampling from normal distributions. In particular, for the ground truth SL pairs, the viability distribution parameters were different between mutated and WT cell lines (see Methods). Subsequently, we compared SLIdR to the Wilcoxon rank sum test and t-test on the simulated data. SLIdR significantly outperformed these tests for predicting SL interactions from simulated data in liver, ovary, and bone cancers (Supplementary Fig. 3). SLIdR was especially advantageous in reducing the number of false positives as illustrated by both the ROC and precision-recall curves, even in rare osteosarcomas with only seven cell lines. The performance difference is more pronounced in the precision-recall curves due to the class imbalance. This imbalance stems from our assumption about the simulated data that the number of true SL pairs is much fewer than non-SL pairs. It is noteworthy that the precision-recall curves are in general more informative for performance comparison in highly imbalanced datasets[46].

## SLIdR recovers experimentally identified SL interactions

Going beyond simulated data, we also evaluated the overlap of SLIdRs pan-cancer predictions with experimentally identified SL interactions from (i) 17 in vitro SL screens reported by Lee et al.[31], and (ii) 10 combinatorial CRISPR screens[17–26]. For these evaluations, we excluded the experimentally identified SL interactions that were not in the set of possible pairwise interactions in the DRIVE dataset, and focused on 978 and 1301 unique experimentally identified SL interactions in the in vitro and CRISPR screens, respectively (see Methods). The retained 1301 unique experimentally identified SL interactions included pairs from 8 of the 10 CRISPR screens. Only four interactions were shared across these 8 combinatorial CRISPR screens. In contrast, SLIdR recovered a significant fraction of established SL interactions (hypergeometric p-values < 10⁻⁹) with sensitivities of 12.3% and 11.45% in the in vitro SL screens and CRISPR screens, respectively. With accuracies of ~93% across both sets, these results comprehensively validated SLIdR predictions (Supplementary Data 4).

Overall, these results highlight the ability of SLIdR to identify well-established and potential targets in both pan-cancer and cancer type-specific settings. Exploring these SL interactions, we observed that the

overlap between pan-cancer and cancer type-specific predictions is limited, re-emphasizing the need to explore the data in both settings, as they provide complementary information. The flexibility of SLIdR to work on both small and large sample sizes makes it an attractive method for this problem.

## SLIdR identified two putative targets in hepatocellular carcinoma

In hepatocellular carcinoma (HCC), we identified nine SL pairs (Fig. 3c). Identifying new potential lethal interactions is particularly relevant for the treatment of liver cancer, where few targeted therapies are available[47]. To demonstrate the predictive power of SLIdR, we sought to validate two of our top hits in HCC, namely *ARID1A-TEAD1* and *AXIN1-URI1* (Fig. 4a and Supplementary Fig. 5a). Both *ARID1A* and *AXIN1* loss-of-function mutations are highly prevalent in HCC patients. *ARID1A* is a SWI/SNF chromatin remodeling gene, and it is mutated in ~9% of hepatocellular carcinomas[48]. *AXIN1* encodes for a key Wnt signaling factor, and it is mutated in 5–15% of HCCs[49]. *TEAD1* and *URI1* have been shown to act as oncogenes and potential therapeutic targets in liver cancer[50–52]. First, we validated the SL interaction between *ARID1A* and *TEAD1* in vitro, using SNU449, a HCC cell line carrying an *ARID1A* somatic mutation. Silencing of *TEAD1* in SNU449 cells significantly impaired cell proliferation compared to control cells (Fig. 4b, c). Using different concentrations of siRNAs, we observed that the phenotype induced by the knock-down of *TEAD1* was dose and time-dependent, thus indicating a specific on-target effect (Fig. 4b, c).

In addition, we inhibited TEAD1 function using an orthogonal method, specifically the small molecule inhibitor verteporfin. The transcriptional activity of TEAD transcription factors relies on their binding with the Yes-associated protein (YAP) coactivator, a well-known effector of the Hippo-signaling pathway[53]. Verteporfin disrupts the YAP-TAZ complex, therefore inhibiting the transcription of the downstream targets[54] (Fig. 4d). Treatment with different dosage of verteporfin significantly impaired cell proliferation and induced cell death in *ARID1A* mutant SNU449 cells (Fig. 4e–g), indicating that the SL interaction between *ARID1A* and *TEAD1* requires *TEAD1* transcriptional activity and is dependent on the Hippo-signaling pathway. Furthermore, to demonstrate that indeed SNU449 sensitivity to verteporfin was dependent on presence of mutant *ARID1A*, we rescued *ARID1A* wild-type expression in SNU449 cells. Indeed, we observed that rescuing wild-type *ARID1A* desensitized cells to verteporfin (Fig. 4h and Supplementary Fig. 6).

Cancer cell lines carry multiple genetic alterations. Therefore, to prove that inhibition of *TEAD1* is specifically synthetic lethal to *ARID1A* loss-of-function, we employed two HCC-derived cell lines carrying a wild-type *ARID1A*–Huh-7 and HLE, and modulated *ARID1A* and *TEAD1* expression using siRNAs (Supplementary Fig. 4a, b). Both Huh-7 and HLE cells transfected with siRNAs targeting both *ARID1A* and *TEAD1* proliferated significantly less compared to control cells or cells silenced for each gene individually (Fig. 5a and Supplementary Fig. 4c). Additionally, inhibition of TEAD1 using the small molecule verteporfin reduced proliferation in both cell lines only upon *ARID1A* silencing, while it did not impact the viability of control cells, thus indicating that loss of *ARID1A* sensitized cells to the treatment (Fig. 5b and Supplementary Fig. 4d). Indeed, *ARID1A*-silenced Huh-7 and HLE cells both showed a lower IC50 for verteporfin compared to control cells (Fig. 5c and Supplementary Fig. 4e). *ARID1A*-silenced cells treated with verteporfin additionally showed a significantly higher proportion of apoptotic cells compared to untreated cells or treated control cells in both Huh7 and HLE cell lines (Fig. 5d, e and Supplementary Fig. 4f).

To assess whether *ARID1A* expression levels would also modulate response to verteporfin in vivo, we employed an in vivo model of the chicken chorioallantoic membrane (CAM), a densely vascularized extraembryonic tissue[55,56]. We treated *ARID1A*-silenced and control Huh-7 cells with verteporfin (1 μM) or vehicle (DMSO) for 24 h. We then

 

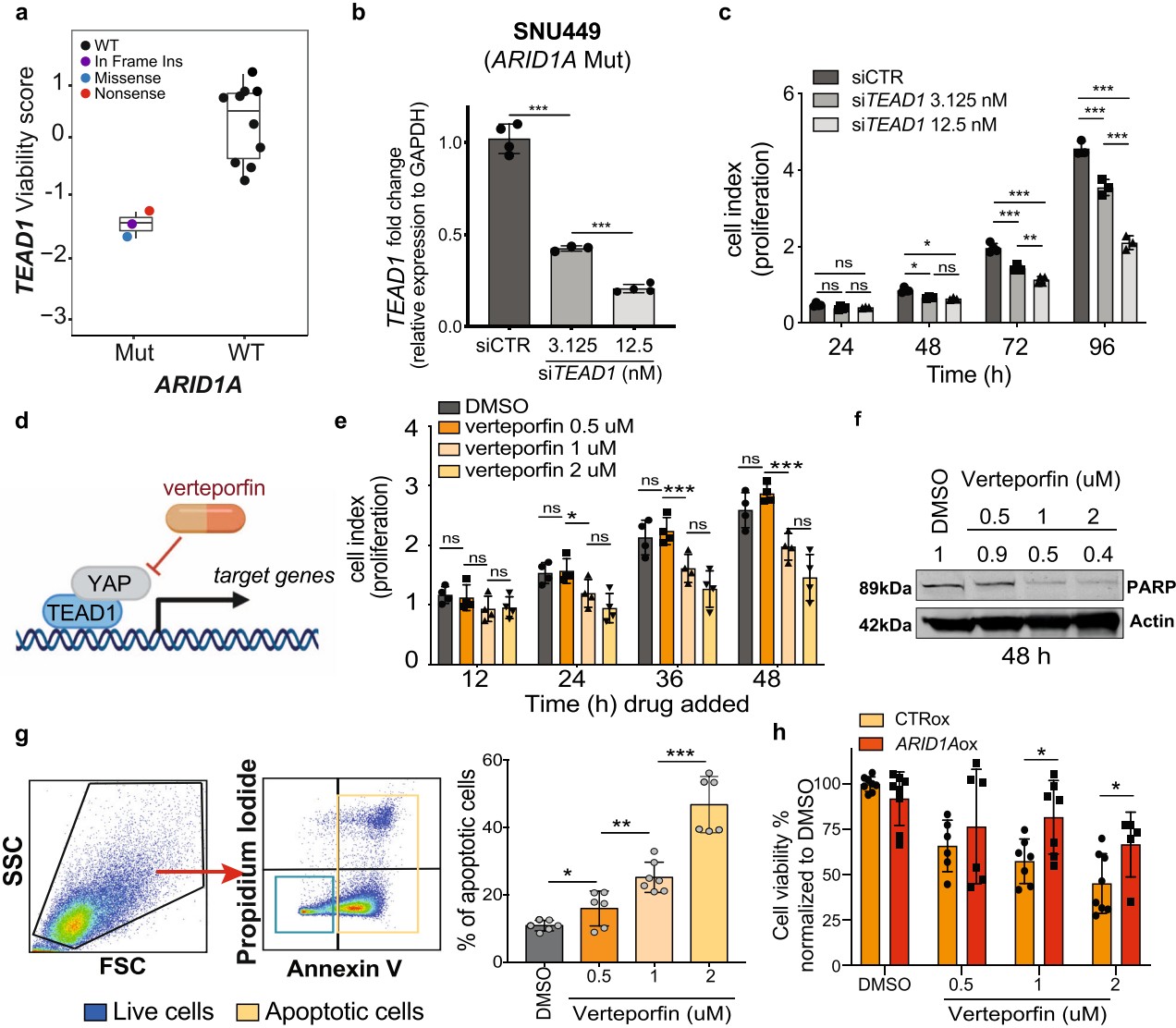

**Fig. 4 | Inhibition of *TEAD1* is deleterious in *ARID1A* mutant liver cancer cells.**
**a** Viability scores of *ARID1A* mutant vs wild-type (WT) HCC cell lines with *TEAD1* knockdown from Project DRIVE dataset, where $n = 13$ HCC cell lines subject to *TEAD1* knockdown experiment. Data are presented as boxplots: Mutant = {min = −1.666, lower (1st Qu.) = −1.5375, middle (median) = −1.409, upper (3rd Qu.) = −1.3190, max = −1.229} and WT = {min = −0.722, lower (1st Qu.) = −0.3455, middle (median) = 0.442, upper (3rd Qu.) = 0.7295, max = 1.044}. **b** RNA expression level (fold-change) of *TEAD1* relative to *GAPDH* in SNU449 cells transfected with control siRNA or with different concentrations (3.125 and 12.5 nM) of *TEAD1* siRNA. RNA levels were assessed by quantitative real-time PCR (qPCR). **c** Proliferation kinetic of SNU449 cells (*ARID1A* mutated) transfected with control siRNA or with different

concentrations (3.125 and 12.5 nM) of *TEAD1* siRNA. **d** Schematic representation of verteporfin mechanism of action. **e, f, g** Proliferation kinetic (**e**), immunoblot for PARP (**f**), and apoptosis assay using Annexin V and propidium iodide (PI) staining (**g**) of SNU449 cells treated with vehicle (DMSO) or different dosage of verteporfin. **f** Protein lysates of SNU449 cells 48 h post-treatment with DMSO or different dosage of verteporfin up to 48 h. **h** SNU449 cells overexpressing ARID1A and control cells treated with vehicle (DMSO) or different dosage of verteporfin. Error bars represent mean (+/−SD) from $n \geq 2$ replicated. For all experiments performed, statistical significance was assessed by two-sided multiple *t*-tests (\**P* < 0.05, \*\**P* < 0.01, \*\*\**P* < 0.001). **d** was generated using BioRender.

---

inoculated the cells into the CAMs and screened the eggs for tumor formation 4 days later (Fig. 5f). In accordance with our in vitro results, verteporfin treatment reduced the volume of tumors formed by *ARID1A*-silenced cells, but not in control cells (Fig. 5g, h), suggesting that *ARID1A* expression modulates response to verteporfin in the CAM model as well.

To further corroborate the predictive power of our method, we additionally validated the SL interaction between *AXIN1* and *URI1*. Similar to the previous SL pair, knockdown of *URI1* affected the viability of *AXIN1*-mutant and *AXIN1*-silenced HCC cells (Supplementary Fig. 5). On the contrary, dual silencing of *AXIN1* and *TP53*, predicted as a non-significant SL pair by SLIdR, did not result in decreased proliferation in HCC cells (Supplementary Fig. 5), thus confirming the

specificity of the synthetic lethal interactions predicted by SLIdR. Taken together, our experimental data strongly support the two HCC SL pairs predicted by SLIdR.

**Integration with CRISPR data**
We applied SLIdR to CRISPR data from the DepMap consortium (Project Achilles)[57] to identify SL pairs. In both pan-cancer and cancer type-specific settings, we reused the mutation data from the Project DRIVE analyses, replacing the viability scores from DRIVE with the CERES[58] viability scores from the CRISPR screens (see Methods). Subsequently, we ran SLIdR in both settings. In the pan-cancer analysis of the CRISPR data, we found 104 SL pairs including *KEAP1-NFE2L2, RPL22-RPL22L1, RPL22-WRN, APC-CTNNB1*, and *MTAP-PRMT5* from the DRIVE

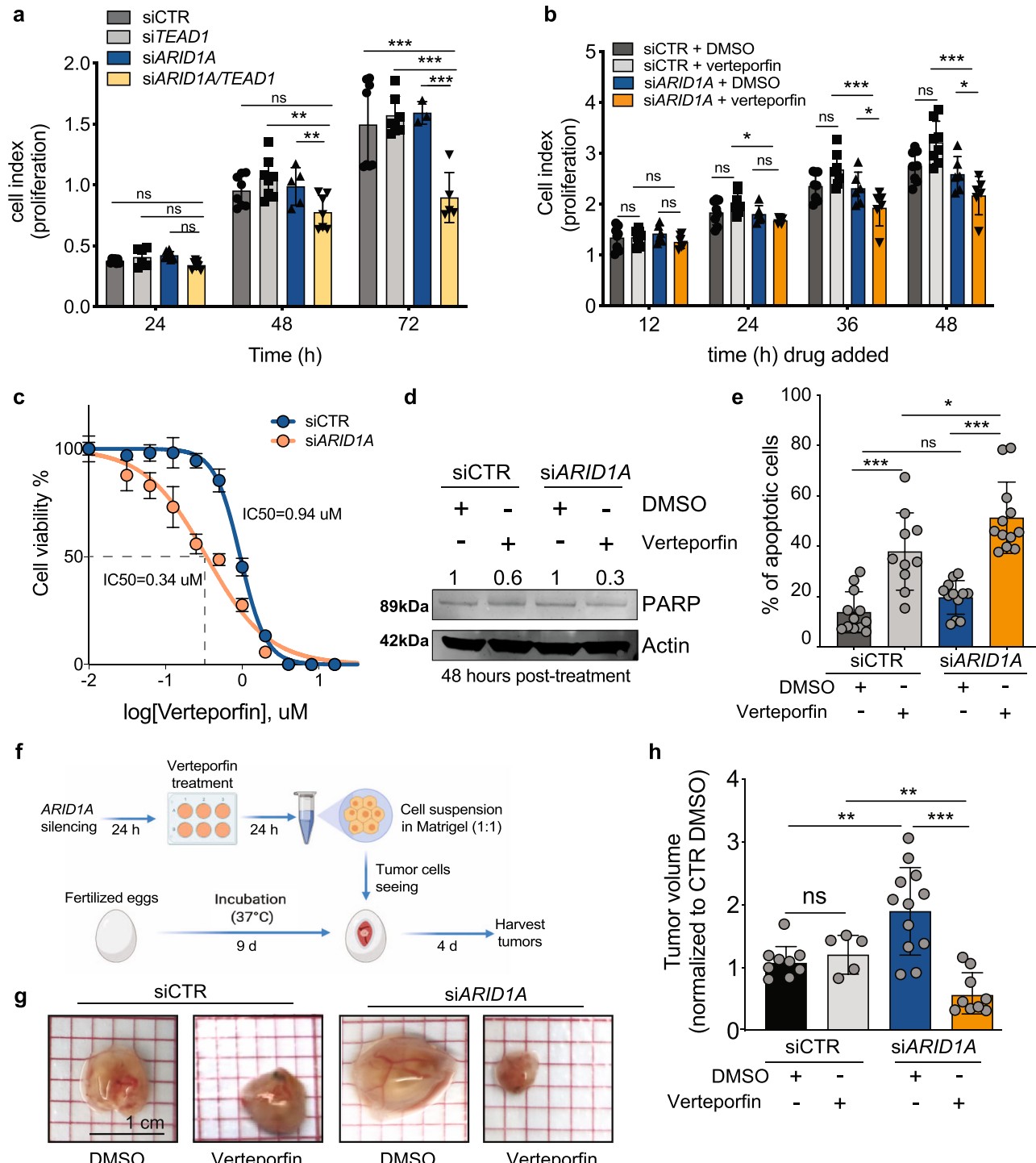

**Fig. 5 | *ARID1A* down-regulation sensitizes liver cancer cells to *TEAD1* inhibition. a** Proliferation kinetic of Huh-7 cells (*ARID1A* wild-type) transfected with control siRNA, *ARID1A* siRNA, *TEAD1* siRNA or both. **b** Proliferation kinetic of Huh-7 cells transfected with control or *ARID1A* siRNA and treated with DMSO or verteporfin (1uM). **c** Dose-response curve of verteporfin in Huh-7 cells transfected with control or *ARID1A* siRNA. **d, e** Immunoblot of PARP (**d**) and apoptosis assay using AnnexinV and propidium iodide (PI) staining (**e**) of Huh-7 cells transfected with control or *ARID1A* siRNA 48 h after treatment with DMSO or verteporfin (1 uM). Immunoblot quantification is relative to loading control (actin). **f** Schematic representation of the CAM assay. **g, h** Representative pictures (**g**) and relative volume quantification (**h**) of tumors explanted from the CAM and derived from Huh-7 cells transfected with control siRNA or *ARID1A* siRNA and treated with DMSO or verteporfin (1uM). Error bars represent mean (+/−SD) from *n* ≥ 2 replicated. For all experiments performed, statistical significance was assessed by two-sided multiple *t*-tests (*$P < 0.05$, **$P < 0.01$, ***$P < 0.001$). **f** was generated using BioRender.

pan-cancer analysis. While the overlap between the two screens was significant (hypergeometric *p*-value < 10⁻¹⁶), we observed that several established and strong candidate pairs of the DRIVE analysis, such as *MTAP-MAT2A*, *MTAP-RIOK1*, and *TP53-TP53BP1*, were missed in the

CRISPR analysis by small margins. This could partly be attributed to the reduced power in the CRISPR analysis, as the data had only 266 cell lines in common with the DRIVE data, i.e., 107 cell lines from the DRIVE screen were missing in the CRISPR screen. Another factor is the

differences in pre-processing and normalization steps across the two screens. Therefore, using Fisher's method to combine the corresponding *p*-values from both screens for each SL pair and retaining only the significant pairs, we identified 162 robust SL pairs across both screens (see Methods). These 162 pairs retained 62% (91 SL pairs) of the original DRIVE candidate SL pairs (Supplementary Data 5), including strong pairs, such as *ACVR2A-WRN, RB1-E2F3,* and *RB1-SKP2,* backed by literature evidence (Supplementary Data 2).

Extending the analysis to the cancer type-specific setting, we found several robust hits in 15 different cancer types (Supplementary Data 5). Some of these were supported by evidence in the literature, including *BRAF-DUSP4, BRAF-MAPK1,* and *BRAF-PEA15* in skin, *GATA3-ESR1* in breast, and *CDH1-CTNNB1* in stomach cancers (Supplementary Data 2). In addition to being common between the two screens, *CD93-ALDH18A1* in pancreatic, *KEAP1-ATP1A1* in lung, *BRAF-MAPK1, BRAF-MDM2,* and *BRAF-CDK5* in skin*, RHOA-CTNNB1* in stomach, and *OR6C1-MDM2* in large-intestine cancers also found support in the PRISM dataset (significance level $\alpha = 0.1$) before multiple testing correction (Supplementary Data 3).

Similar to the pan-cancer setting, using Fisher's method, we were able to leverage the complementary information across the two screens in the cancer type-specific setting. The method proved especially beneficial in retaining pairs which showed reduced signal or were dropped due to the *p*-value threshold for WT cell lines in one of the two screens. Some examples of such robust hits include *KEAP1-TAPT1* in lung cancer - a dependency recently established by Romero et al.[59], *EIF3C-MTOR* in pancreatic cancer (Supplementary Data 2, 3), *ARID1A-TEAD1, ARID1A-FERMT2,* and *AXIN1-URI1* in hepatocellular carcinoma (Figs. 3c, 4, 5; Supplementary Figs. 4, 5), and *GATA3-MDM2* in breast cancer. In a recent study by Bianco et al.[60], *GATA3* and *MDM2* were found to be synthetic lethal in estrogen receptor-positive breast cancers. Thus, through parallel analysis and integration of the two datasets, SLIdR reports robust and complementary SL pairs.

## Discussion

With precision medicine advancing in oncology, there is a need to identify mutation-specific therapeutic options for different types of cancer. Most current methods for predicting novel drug targets perform pan-cancer multi-omics analysis and use genetic perturbation screening data, if at all, merely to refine the candidate list of SL pairs. To make better use of these data and generate improved predictions, we developed SLIdR, a rank-based statistical method for predicting mutation-specific SL partners from large-scale perturbation screens.

Following the work of McDonald et al.[27], we initially applied SLIdR to the Project DRIVE dataset in the pan-cancer setting and recovered most of the SL interactions reported by them in addition to several candidate pairs. However, further analysis of these pan-cancer hits revealed that the signal for many pairs were highly specific to a particular cancer type, warranting the need to identify cancer type-specific SL pairs. Thus, we applied SLIdR to each cancer type individually and identified several well-known as well as candidate pairs. Through a detailed literature survey, systematic validation on large-scale drug-response profiles, and integration with CRISPR screening data, we found strong supporting evidence for several pairs in both the pan-cancer and cancer type-specific settings. These lists of candidate targets, while still requiring extensive validation in follow-up experiments, hold promise for drug repurposing and designing targeted therapies.

A key advantage of SLIdR is its ability to predict SL pairs even with small sample sizes while reducing the number of false positive targets. SLIdR improves the statistical power by using statistical tests based on the Irwin-Hall distribution on the ranked viabilities across all genes. In a simulation study, we have illustrated this advantage and superiority of SLIdR by comparing it to the Wilcoxon rank sum test adopted by ISLE and DAISY. In addition, SLIdR recovered experimentally identified SL

interactions from in vitro SL screens and CRISPR screens with ~93% specificity. Thus, we propose that by replacing the Wilcoxon rank sum test step with SLIdR, methods such as ISLE and DAISY could reduce the number of false positive candidate SL pairs in their initial sets and improve their overall performance.

To corroborate the predictive power of our method, we experimentally validated two synthetic lethal interactions—*ARID1A-TEAD1* and *AXIN1-URI1*, in hepatocellular carcinoma. Specifically, we proved that inhibition of *TEAD1* function, using siRNA or the TEAD-YAP inhibitor verteporfin, is lethal in *ARID1A*-mutated or *ARID1A*-silenced HCC cell lines. We additionally showed that treatment with verteporfin significantly reduced the growth of *ARID1A*-silenced HCC cells in vivo. Taken together, our results show that *ARID1A* and *TEAD1* are synthetic lethal interactors in HCC and strongly indicate that this relationship is dependent on the regulatory function of *TEAD1* in the Hippo-signaling pathway. Indeed, our results are in accordance with the recent work of Chang et al.[61]. Given the widespread role of the SWI/SNF complex[62], the frequency of *ARID1A* inactivation in several malignancies other than HCC[62], and the availability of TEAD-YAP inhibitors[53], the identification of the *ARID1A-TEAD1* synthetic lethal pair provides an example of how SLIdR can help improve cancer therapy. We also validated another independent synthetic lethal pair, *AXIN1-URI1*, in HCC, further corroborating the predictive power of our method.

While SLIdR was successful in identifying SL pairs, there are several limitations: (1) In its current scope, SLIdR primarily focuses on identifying SL partners from perturbation screening data, copy number, and mutation data. Consequently, it failed to recover the predictions based on expression and pathway reported by McDonald et al.[27], emphasizing the importance of integrating other omics data. Thus, extending SLIdR to incorporate multi-omics and pathway data or using it in tandem with methods like ISLE could further improve its overall performance. (2) Occasionally, the results from SLIdR are sensitive to the WT *p*-value threshold, resulting in losing important SL pairs. While this threshold is important for filtering out candidates showing increased or decreased viabilities in WT cell lines, the choice of the threshold depends on the data type and processing steps, requiring the user to optimize this threshold to their specific setting. For improved versatility, an extension of SLIdR could include a score to prioritize the pairs instead of filtering based on the WT *p*-value threshold. (3) As cancer cells can harbor multiple mutations, SLIdR initially identified the same SL partner for different mutations mapping to the same cells, especially in the pan-cancer setting. To overcome this issue and identify the true SL pairs from several potentially confounding mutated genes, we used propensity score matching-based potential outcome models and identified significant SL pairs. Propensity-score matching tries to replicate a randomized experiment by matching samples between mutated and wild-type groups on other confounding mutations, thereby obtaining balanced covariate distributions across the two groups. Consequently, the average causal effect using matching is determined only from a subset of data. Since the mutation matrix is sparse, matching yields under-powered subsets in certain cases, inadequate for robust estimation of the average causal effect. Furthermore, our current approach for controlling confounding drivers is prone to selection bias, because we do not account for the underlying dependency structure between confounding drivers due to small sample sizes. Further research is required to improve the estimation of the causal effect of driver mutations from sparse mutation data. A possible solution would be to use information from other omics data to select the confounders to adjust for, thus improving the overall estimate.

Taken together, SLIdR provides a robust statistical framework for rapid discovery of SL interactions from large-scale perturbation screens in both pan-cancer and cancer type-specific settings. Particularly in precision oncology, SLIdR can help in developing putative mutation-specific and effective personalized therapies.

## Methods

### Primary screening data

We used viability profiles published in Project DRIVE[27], as well as corresponding mutation data and copy number data from the Cancer Cell Line Encyclopedia (CCLE) collection[28] for 373 cell lines across 23 cancer types.

### Viability data from Project DRIVE

Viability data captures the viability of cell lines for each gene knockdown experiment. In Project DRIVE, 7837 genes were targeted by using an average of 20 pooled shRNAs per gene. The shRNA activities were defined as the quantile normalized log fold change in shRNA read counts 14 days after the start of the knockdown experiment to the shRNA abundance in the input library. The gene-level viability score of each cell line was computed by aggregation of shRNA activities using two computational methods, namely RSA[63] and ATARiS[64]. The RSA method uses all shRNA reagents targeting a gene and can be used to identify essential, inert, and active genes, while ATARiS only uses a subset of shRNAs with consistent activity across the cell lines and aims to provide a robust gene-level score by discarding shRNA reagents with off-target effects. ATARiS provides a relative score for the gene-level activity by median-centering the data for each reagent and, as a result, cannot distinguish between inert and essential genes.

To process the viability data, we removed essential genes using the RSA method as was performed in Project DRIVE[27]. Genes with an RSA value ≤ −3 in more than 50% of cancer cell lines were reported as essential genes. In total, 460 and 185 genes were reported essential in cancer type-specific and pan-cancer settings, respectively. The resulting viability matrices consisted of the ATARiS scores for the remaining perturbed genes (rows) for each cell line (columns).

### Mutation and copy number data

For the pan-cancer setting, we focused on genes with mutations or copy number aberrations in more than 30 cell lines. We downloaded mutation data and copy number data from the CCLE website, and binarized them as follows. A gene in a given cell line was assigned a value of 1 if it was subject to non-synonymous mutations, and 0 otherwise. For copy number data, we focused only on deep deletions and binarized a gene in a given cell line by assigning a value of 1 if the gene was homozygously deleted and a value of 0 otherwise. Finally, combining both these data, a driver gene in a given cell line was assigned a value of 1 if it was subject to non-synonymous mutations, deep deletions, or both; and 0 otherwise.

In the cancer type-specific setting, to define the set of driver genes, we first used the MutSig 2CV v3.1[32,33] MAF file from TCGA for each cancer type and focused only on significantly mutated genes ($q$-value ≤ 0.05). Next, we concentrated on genes with non-synonymous mutations in two or more cell lines, and excluded copy number data as it was very noisy in this setting. Thus, a gene in a given cell line was assigned a value of 1 if it was subject to non-synonymous mutations, and 0 otherwise. The resulting binarized mutation matrices described the mutation profiles for each cell line (column) across all driver genes (rows).

### SLIdR algorithm

SLIdR is a rank-based statistical framework to identify the presence of synthetic lethal dependency between a driver gene $d$ and a perturbed gene $g$. For each driver gene $d$, we divided the cell lines into two groups according to the mutation status of $d$, namely wild-type cell lines (WT) and mutated cell lines (Mut). Further, we ranked the perturbed genes by their ATARiS scores, for each mutated and WT cell line, and normalized it between 0 and 1. Due to a large number of perturbed genes, the normalized ranks have many distinct levels and are highly fine-grained. Hence, we assumed the normalized ranks to be continuous.

Let $(d, g)$ be a fixed pair of driver and perturbed gene and $C_d$ be the set of cell lines mutated in $d$ of cardinality $n$. If $(d, g)$ is an SL pair, based on the aforementioned definition (Fig. 1a), a mutation in driver gene $d$ in combination with knockdown of gene $g$, results in low viabilities in mutated cell lines $C_d$. We used a one-sided statistical test based on the Irwin-Hall distribution to test whether the viabilities of mutated cell lines $C_d$ from knockdown of gene $g$ are lower than expected by chance. We defined the null hypothesis $H_0$ as the knockdown of gene $g$ having no impact on the viability of the cell lines in $C_d$. For each cell line $c \in C_d$, we computed the normalized rank of the viability of $c$ from knockdown of gene $g$ across all other gene knockdowns in cell line $c$, and denoted this rank as $r_{c|g}$. Under the null hypothesis, the normalized ranks take uniform random values in the interval [0, 1], $r_{c|g} \sim U(0, 1)$. The test statistic $T$ for the pair $(d, g)$ is then defined as the sum of normalized viability ranks of mutated cell lines $C_d$ perturbed in gene $g$, $T = \sum_{c \in C_d} r_{c|g}$. Under $H_0$, the test statistic $T$ is the sum of $n$ independent uniform random variables on the unit interval and hence it follows the Irwin-Hall distribution of order $n$. The resulting $p$-value was computed as the lower tail probability $P(T < t_{obs})$, where $t_{obs}$ is the observed test statistic. For large $n$, computation of the Irwin-Hall probability distribution is either computationally expensive or numerically unstable. Therefore, we used the approximation $T \sim N(n/2, n/12)$ for $n > 20$.

Conversely, based on the definition of synthetic lethality (Fig. 1a), wild-type cell lines with respect to driver gene $d$ are expected to behave similar to healthy cells when perturbed in gene $g$. Therefore, it is important to filter out genes which upon knockdown adversely alter the viabilities of WT cell lines. We used a two-sided Irwin-Hall test to filter out any pair $(d, g)$ that reached statistical significance ($\alpha = 0.1$) in the WT cell lines. However, we did not use this filter for the pan-cancer setting due to the diverse nature of the cell types and cancer types. Since knockdown data of several driver genes is unavailable in the Project DRIVE screen, we tested for SL interactions only in one direction and were unable to test the effect on viability of the cell lines when the driver genes alone are knocked down.

### Multiple testing

We reduced the number of false positives arising from multiple testing by choosing a significance level of $1/(M \times N)$, where $M$ is the number of knockdowns and $N$ is the number of driver genes. Therefore, we expect on average one false positive hit among all reported SL hits, for each cancer. Our method is computationally inexpensive as it avoids performing all $M \times N$ tests. For each driver, we compute the test statistic for all perturbed genes and sort them in ascending order. The pre-ordering of the test statistics enables us to test for genes until the corresponding $p$-value is less than the chosen significance level. Further, we note that this approach was in good agreement with controlling the false discovery rate at 10% (Supplementary Fig. 1c).

### Causal inference

Cell lines are often subject to mutations or aberrations in multiple driver genes and, as a result, different driver genes pair with the same SL partner gene (Fig. 2b; Supplementary Fig. 1a). This is typically not an issue in the cancer type-specific setting but is prevalent in the pan-cancer setting. In order to identify the most likely SL pairs from the many confounding driver genes, we used matching-based potential outcome models. The main goal of the matching method is to emulate a randomized experiment by matching samples of treated and control groups according to covariates, thereby obtaining similar covariate distributions across the two groups.

Let $S = \{d_1, \ldots, d_k\}$ be a set of $k$ driver genes pairing with the same SL partner gene $g$. For a given driver gene $d_i \in S$, the cell lines mutated and wild-type in $d_i$ constitute the treated and control groups, respectively, and the viability from knockdown of the SL partner gene $g$ is used as the response or outcome variable. $S_{-d_i} = S \setminus \{d_i\}$ is the set of all the driver genes in $S$ excluding $d_i$ and constitutes the set of

confounding covariates. We used the *Matching* R package[65] and performed propensity-score matching with a caliper of 0.1 to match the treated and control groups based on confounding covariates, i.e., the mutation status of all the driver genes in $S_{-d_i}$. Since matching is dependent on the order of the samples, we reshuffled and repeated matching 50 times. After each run, we recorded the standardized mean difference (smd) between the two groups for all covariates and chose the run with the lowest sum of smd across all covariates. Finally, for the chosen run, we performed a paired t-test between the responses of treated and control groups. We repeated these steps for all $k$ driver genes in $S$.

We repeated the above described process for all such sets of driver genes pairing with the same SL partner gene and reported the pairs that reached statistical significance ($\alpha = 0.05$) in the paired t-test (Fig. 2c; Supplementary Fig. S1b).

## Simulation Study

We performed a simulation study focusing on both prevalent and rare cancer types, including liver, ovarian, and bone cancers with 13, 14, and 7 cell lines, respectively. For each cancer type, we reused the binarized mutation matrix from the cancer type-specific analysis on the Project DRIVE dataset and simulated the corresponding viabilities. For a given cancer type, the ground truth comprised 30 random pairwise interactions between driver genes and perturbed genes. The viability matrix for each cancer type was simulated in two steps. First, the viabilities of all the cell lines across all perturbations were simulated by sampling from $N(\mu_s, \sigma_s^2)$, where $\mu_s$ and $\sigma_s$ are the mean and standard deviation of viabilities from the Project DRIVE data for the primary site $s$, respectively. Next, for each true SL pair $(d, g)$, we stratified the cell lines into WT and mutated based on the mutation status of the driver gene $d$. Since true SL pairs should exhibit differential viability profiles between mutated and WT cell lines, the viabilities of mutated cell lines from perturbation of the true SL partner gene $g$ were sampled from another distribution $N(\mu_s^{mut}, \sigma_s^{mut2})$, where $\mu_s^{mut} = (min_s + \mu_s)/2$ and $\sigma_s^{mut} = 1.2 * \sigma_s$, and $min_s$ is the minimum of viabilities from the Project DRIVE data for the primary site $s$. Using these parameters ensured that (i) the perturbation of gene $g$ resulted in reduced viabilities in mutated cell lines than in WT cell lines, and (ii) the simulated viabilities were congruent with the Project DRIVE data. This was repeated for all 30 true SL pairs in each cancer type.

Finally, we applied SLIdR to the simulated data and compared it to Wilcoxon rank sum test and t-test applied to the raw simulated viabilities. For each possible pairwise interaction in a given cancer type, we stratified the cell lines into mutated and WT cell lines based on the mutation status of the driver gene constituting the pair. Then, we tested whether the raw viabilities of the mutated cell lines from the perturbation of the partner gene was less than zero using a one-sided Wilcoxon rank sum test and t-test. The resulting p-values from SLIdR, Wilcoxon rank sum test, and t-test were compared to the true labels in order to plot the ROC and Precision-Recall curves. The true labels comprised a binary vector of all possible pairwise interactions with 1 assigned to true SL pairs and 0 otherwise. This was repeated for all three cancer types.

## Comparison of experimentally identified SL interactions

To compare SLIdRs predictions with established SL interactions, we focused on (i) 6,033 experimentally identified SL interactions from 17 in vitro focused SL screens reported by Lee et al.[31], and (ii) 24,651 experimentally identified SL interactions from 10 combinatorial CRISPR screens[17–26] (details in Supplementary Note 1). Since we applied SLIdR to the Project DRIVE dataset, we excluded the experimentally identified SL interactions that were not in the set of possible pairwise interactions in the DRIVE dataset (9,443,304), yielding 978 and 1301 unique experimentally identified SL interactions in the in vitro screens and CRISPR screens, respectively. The retained 1301 unique

experimentally identified SL interactions included pairs from 8 of the 10 CRISPR screens and four shared SL interactions (Supplementary Data 4). Finally, we re-ran SLIdR in the pan-cancer setting by relaxing the significance level to 5% and used a hypergeometric test to assess whether the overlap between predicted and experimentally identified SL pairs was significant. We also computed the sensitivity, specificity, and accuracy in both sets.

## Validation on PRISM dataset

We validated the predicted pan-cancer and cancer type-specific SL pairs on the primary PRISM drug-response screen containing the growth inhibitory activity of 4686 drugs tested across 578 human cancer cell lines[37]. For both settings, we first filtered out all the pairs exhibiting oncogene-addiction and retained only those pairs with a targetable SL partner gene from the drugs in the screen. As previously described in the SLIdR algorithm, for each predicted SL pair, we stratified the cell lines based on the mutation status of the driver gene into WT cell lines and mutated cell lines. Subsequently, we tested if the drug compound targeting the corresponding SL partner gene resulted in reduced viabilities of mutated cell lines in comparison to WT cell lines using a one-sided t-test. Since multiple drug compounds could target the same SL partner gene, for a given SL pair we tested the differential drug response for each drug compound. We repeated this for all SL pairs and reported the SL pairs with a significant difference in drug response at a significance level of $\alpha = 0.1$ along with their multiple testing corrected q-values.

## Control experiments on PRISM screen

To assess the robustness of the PRISM validation results of SLIdR hits, we compared them to the PRISM results obtained from 1000 sets of random gene pairs. The same process was followed for both pan-cancer and cancer type-specific settings. We first retained only those perturbed genes with at least one matching drug compound in the PRISM screen for each cancer. We then generated a set of all possible pairwise interactions ($U$) between all the driver genes and druggable perturbed genes and filtered out all the pairs exhibiting oncogene-addiction. Subsequently, we ran permutation tests on random sets of gene pairs. For each run, we sampled $K$ gene pairs from $U$ at random, where $K$ is the number of SL pairs predicted by SLIdR and tested on PRISM screens. For each sampled gene pair, we stratified the cell lines based on the mutation status of the driver gene into WT and mutated cell lines. Then, we tested whether the drug compound targeting the corresponding partner gene reduced the viabilities of mutated cell lines compared to WT cell lines using a one-sided t-test. Since multiple drug compounds could target the same partner gene, we tested the differential drug response for each drug compound for a given gene pair. We repeated this for all random gene pairs and counted the gene pairs with a significant difference in drug response (significance level of $\alpha = 0.1$). Finally, we computed the empirical p-values by comparing the number of significant drug responses of SLIdR hits and those of random sets.

## SLIdR on CRISPR dataset

With Project DRIVE as our primary screen, we extended SLIdR to CRISPR data from the DepMap consortium (Project Achilles 20Q2)[57] to identify robust hits. This secondary screen consisted of CERES[58] viability scores of 769 cell lines across 18,119 CRISPR knockout experiments. However, only 266 cell lines were common with the primary screen and were subsequently used for the downstream analysis.

In both pan-cancer and cancer type-specific settings, we reused the mutation data from the Project DRIVE analyses, replacing only the viability scores from DRIVE with the CRISPR viability scores. However, it should be noted that the reduced overlap of cell lines in the CRISPR screen resulted in dropping a few driver genes in a few cancer types. Specifically, a few driver genes were dropped as they lacked non-

synonymous mutations in at least two cell lines in the cancer type-specific setting. For the viability matrices, we focused only on the common perturbations between both the screens across common cell lines. Subsequently, the CRISPR viability scores were median centered across cell lines.

We re-ran SLIdR with the same threshold parameters as with Project DRIVE. To identify the robust hits, we took the union of the sets of SL pairs predicted by SLIdR on both the screens. For the SL pairs unique to the DRIVE screen, we determined the significance of the pair in the CRISPR screen, and vice-versa. Consequently, we combined the $p$-values corresponding to the mutated cell lines from the two screens using Fisher's method, and filtered those with combined $p$-value $< 1/(M \times N)$, where $M$ is the number of knockdowns and $N$ is the number of driver genes in the Project DRIVE analysis. Finally, we filtered out all the candidate pairs that reached statistical significance ($\alpha = 0.1$) in the WT cell lines in both screens. However, we did not use this final filter for the pan-cancer setting due to the diverse nature of the cell types and cancer types.

## Cell lines maintenance

Liver cancer-derived cell lines SNU449, Huh-7, and HLE were obtained from the Laboratory of Experimental Carcinogenesis (Bethesda, MD, USA), authenticated by short tandem repeat profiling as previously described[66], and tested for mycoplasma infection using a PCR-based test (ATCC). All cell lines were maintained under the conditions recommended by the provider. Briefly, all cell lines were cultured in DMEM supplemented with 5% Fetal Bovine Serum (FBS), non-essential amino-acids (NEAA) and antibiotics (Penicillin/Streptomycin). The cells were incubated at 37 °C in a humidified atmosphere containing 5% $CO_2$. Exponentially growing cells were used for all in vitro studies.

## Transient gene knockdown by siRNAs

Transient gene knockdown was conducted using ON-TARGET plus siRNA transfection. ON-TARGET plus SMARTpool siRNAs against human *TEAD1*, *ARID1A*, *URI1*, *AXIN1* and *TP53*, ON-TARGET plus SMARTpool non-targeting control, and DharmaFECT reagent were all purchased from GE Dharmacon (Supplementary Data 6). Transfection was performed according to the manufacturer's protocol. Briefly, log-phase liver cancer cells were seeded at ~60% confluence. Since residual serum affects the knockdown efficiency of ON-TARGET plus siRNAs, growth medium was removed as much as possible and replaced by serum-free medium (Opti-MEM; Supplementary Data 6). siRNAs were added to a final concentration of 25 nM. siRNAs targeting different genes can be multiplexed. Cells were incubated at 37 °C in 5% $CO_2$ for 24–48–72 h (for mRNA analysis) or for 48–72 h (for protein analysis). In order to avoid cytotoxicity, the transfection medium was replaced with a complete medium after 24 h.

## Transient *ARID1A* overexpression

For *ARID1A* overexpression, pcDNA6-ARID1A (#39311)[67] was obtained from Addgene (https://www.addgene.org), and the empty control vector was pCDNA6-V5/His.b (ThermoFisher Scientific). The expression vectors were transiently transfected using the Lipofectamine 3000 transfection reagent (ThermoFisher Scientific) following the manufacturer's instructions.

## RNA extraction and relative expression by qRT-PCR

Total RNA and proteins were extracted from cells at 75% confluence using TRIZOL (Supplementary Data 6) according to manufacturer's guidelines. cDNA was synthesized from 1 μg of total RNA using SuperScript™ VILO™ cDNA Synthesis Kit (Supplementary Data 6). All reverse transcriptase reactions, including no-template controls, were run on an Applied Biosystem 7900HT thermocycler. Gene expression was assessed by using FastStart Universal SYBR Green Master Mix (Supplementary Data 6) and all qPCR were performed at 50 °C for

2 min, 95 °C for 10 min, and then 40 cycles of 95 °C for 15 s and 60 °C for 1 min on a QuantStudio 3 Real-Time PCR System (Applied Biosystems). The specificity of the reaction was verified by melting curve analysis. Measurements were normalized using GAPDH level as the reference. The fold change in gene expression was calculated using the standard ΔΔCt method as previously described[68]. All samples were analyzed in triplicates.

## Proliferation assay

Cell proliferation was assayed using the xCELLigence system (RTCA, ACEA Biosciences, San Diego, CA, USA). Background impedance of the xCELLigence system was measured for 12 s using 50 μl of room temperature cell culture media in each well of E-plate 16. Cells were grown and expanded in tissue culture flasks as previously described. After reaching 75% confluence, the cells were washed with PBS and detached from the flasks using a short treatment with trypsin/EDTA. 5000 cells were dispensed into each well of an E-plate 16. Growth and proliferation of the cells were monitored every 15 min up to 120 h via the incorporated sensor electrode arrays of the xCELLigence system, using the RTCA-integrated software according to the manufacturer's parameters. In the case of transient siRNA transfection, cells were detached and plated on xCELLigence 24 h post-transfection.

## Drug treatment

$5 \times 10^3$ exponentially growing cells were plated in a 96-well plate. After 24 h, cells were treated with serial dilution of verteporfin (Supplementary Data 6) or dimethyl sulfoxide (DMSO). DMSO served as the drug vehicle, and its final concentration was no >0.1%. Cell viability was measured after 72 h using CellTiter-Glo Luminescent Cell Viability Assay reagent (Supplementary Data 6). Luminescence was measured on Varioskan Microplate Reader (ThermoFisher Scientific). Results were normalized to DMSO control. All experiments were performed in triplicates. Results are shown as mean ± SD. Curve fitting was performed using Prism (GraphPad) software and the nonlinear regression equation.

## Apoptosis analysis by flow cytometry

Cells were collected 72 h post siRNA transfection, stained with annexinV (FITC conjugate; Supplementary Data 6) and propidium iodide (PI), and analyzed by flow cytometry using the BD FACSCanto II cytometer (BD Biosciences, USA). Briefly, cells were harvested after the incubation period and washed twice by centrifugation (1200 g, 5 min) in cold phosphate-buffered saline (DPBS; Supplementary Data 6). After washing, cells were resuspended in 0.1 mL AnnV binding buffer 1X (ABB 5X, Supplementary Data 6) containing fluorochrome-conjugated AnnV and PI (PI to a final concentration of 1 ug/mL) and incubated in darkness at room temperature for 15 min. Following immediately, cells were analyzed by flow cytometry, measuring the fluorescence emission at 530 nm and >575 nm. Data were analyzed by FlowJo software version 10.5.3 (https://www.flowjo.com/).

## Immunoblot

Total proteins were extracted by directly lysing the cells in Co-IP lysis buffer (100 mmol/L NaCl, 50 mmol/L Tris pH 7.5, 1 mmol/L EDTA, 0.1% Triton X-100) supplemented with protease and phosphatase inhibitors. Cell lysates were then treated with a reducing agent, boiled and loaded onto neutral pH, pre-cast, discontinuous SDS-PAGE mini-gel system. After electrophoresis, proteins were transferred to nitrocellulose membranes using the Trans-Blot Turbo Transfer System (Bio-Rad). The transblotted membranes were blocked for 1 h in TBST 3% SureBlock (LubioScience) and then probed with appropriate primary antibodies (1:1000) overnight at 4 °C. List of antibodies and working concentrations are available in (Supplementary Data 6). Next, the membranes were incubated for 1 h at room temperature with fluorescent secondary goat anti-mouse (IRDye 680) or anti-rabbit (IRDye

800) antibodies (both from LI-COR Biosciences). Blots were scanned using the Odyssey Infrared Imaging System (LI-COR Biosciences) and band intensity was quantified using ImageJ software. The ratio of proteins of interest/loading control in verteporfin-treated samples were normalized to their DMSO-treated control counterparts. All experiments were performed and analyzed in duplicates.

### Chorioallantoic membrane (CAM)

Fertilized chicken eggs were obtained from Gepro Geflügelzucht AG at day 1 of gestation and were maintained at 37 °C in a humidified (60%) incubator for 9 days[69]. At this time, an artificial air sac was formed using the following procedure: a small hole was drilled through the eggshell into the air sac and a second hole near the allantoic vein that penetrates the eggshell membrane. Mild vacuum was applied to the hole over the air sac in order to drop the CAM. Subsequently, a square 1 cm window encompassing the hole near the allantoic vein was cut to expose the underlying CAM[69]. After the artificial air sac was formed, Huh-7 cells growing in tissue culture were inoculated on CAMs at $1.5 \times 10^6$ cells per CAM, on three to six CAMs each. Specifically, 48 h post-siRNA transfection, *ARID1A*-silenced and control Huh-7 cells were treated with verteporfin (1 µM). 24 h post-treatment, cells were detached from the culture dish with Trypsin, counted, suspended in 20 µl of medium (DMEM) and mixed with an equal volume of Matrigel. To prevent leaking and spreading of cells, an 8 mm (inner diameter) sterile teflon ring (removed from 1.8 ml freezing vials, Nunc, Denmark) was placed on the CAMs and the final mixture was grafted onto the chorioallantoic membranes inoculating the cells with a pipette inside the ring[70]. Embryos were maintained at 37 °C for 4 days after which tumors at the site of inoculation were excised using surgical forceps. Images of each tumor were acquired with a Canon EOS 1100D digital camera. Surface measurements were performed by averaging the volume (height*width*width) of each tumor using ImageJ, as previously described[71]. Total $n \geq 5$ tumors for each condition were analyzed over three independent experiments. Experiments in fertilized eggs were performed before the third period of the embryonation period (before day 14). Experiments in fertilized eggs in this period do not require ethical approval.

### Statistics and reproducibility

No statistical method was used to predetermine sample size. For the in vivo experiments, CAMs were allocated randomly to each condition. For the in vivo model, CAMs were screened for tumor formation blindly by two independent scientists.

For the functional experiment reported in Figs. 4, 5 and Supplementary Figs. 4, 5, 6, statistical analyses were conducted using Prism software v8.0 (GraphPad Software, La Jolla, CA, USA). For in vitro experimental studies, statistical significance was determined by the two-tailed unpaired Student's *t*-test. For comparison involving multiple time points, statistical significance was determined by multiple Student's *t*-test corrected for multiple comparisons with the Holm–Sidak method. A $p$-value < 0.05 was considered statistically significant. For all figures, 'ns' indicates not reaching the significance level. For the CAM assay, a two-tailed unpaired Student's *t*-test was used. Unless otherwise indicated, all data represent the mean ± standard deviation from at least three independent experiments.

For details on the reagents used, please refer to Supplementary Data 6.

### Reporting summary

Further information on research design is available in the Nature Portfolio Reporting Summary linked to this article.

## Data availability

The raw shRNA data has already been published as a part of Project DRIVE (https://data.mendeley.com/datasets/y3ds55n88r/4) and all the mutation and copy number data from CCLE is available at https://portals.broadinstitute.org/ccle. The MutSig 2CV v3.1[32,33] MAF file for each cancer type is available at http://firebrowse.org/. The processed project DRIVE data for running SLIdR in pan-cancer and cancer type-specific settings are available at https://doi.org/10.6084/m9.figshare.21508065.v4[72]. The PRISM drug-response dataset and the CRISPR dataset (Project Achilles 20Q2) are available at https://depmap.org/portal/download/. All these datasets are publicly available. The validation experimental data generated in this study are provided in the Source data file. Source data are provided with this paper.

## Code availability

The latest version of SLIdR package is available at https://github.com/cbg-ethz/slidr[73] along with the scripts used to process and generate the results for the paper. Other R packages used, include, Matching (v4.9-3), tableone (v0.9.3), dplyr (v0.7.6), ggplot2 (v3.2.0.9), cowplot (v0.9.2), circlize (v0.4.5). For the validation experiments the following softwares were used—BD, FACSDiva and FlowJo (10.5.3), Prism for MacOS v8.2.1.

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

## Acknowledgements

This work was partly supported by ERC Synergy Grant 609883 to N.B., by the Swiss Cancer League [KFS-4543-08-2018 to C.K.Y.N. and KFS-4988-02-2020-R to S.P.], and by The Professor Dr Max Cloëtta Foundation (Medical research position) to S.P.

## Author contributions

S.S. and N.B. conceived the presented idea. S.S. and H.M. developed and implemented the method and built the R package. N.B. supervised the computational study. S.P. and C.K.Y.N. supervised the experimental study. S.S. performed the downstream bioinformatics analyses and contributed to the visualizations. G.B., M.C.-L. and M.M. performed in vitro and in vivo experiments. N.B., S.P. and C.K.Y.N. critically discussed the results. S.S., H.M., G.B., M.C.-L., C.K.Y.N., S.P., and N.B. interpreted the results and wrote the manuscript. All authors agreed to the final version of the manuscript.

## Competing interests

The authors declare no competing interests.
