## [Peer Review File · Nature Communications]

Discovery of synthetic lethal interactions from large-scale pan-cancer perturbation screensReviewer #1 (Remarks to the Author): Expertise in SL screens

General comments to the Authors

The authors have developed a method, SLIdR (Synthetic Lethal Identification in R), to identify synthetic lethal (SL) pairs from shRNA genetic screens in cancer cell lines using a novel method and provided an in vitro experimental validation of one predicted SL pair.

The general problem of identifying SL pairs using in both pan-cancer and in a cancer-specific pattern is an important problem. However, in our opinion, this work misses systematic testing of model performance and comparison with available tools, making it very hard to conclude the efficacy of the method and whether it is useful for the community. It also lacks a proper systematic validation of many of the predicted SL pairs. One useful contribution of this work is the specific SL pair (between AXIN1 & URI1) which has been validated in vitro, but for a journal of the level of Nat. Comm. One would expect to see further confirmation in vivo.

Major Comments

1. In the last few years, multiple tools have been published to identify SLs [PMID: ISLE-29959327, DAISY- 25171417, MiSL- 28561042], including a few in Nature Communication. Tools like MiSL (published in Nature Comm.) identifies mutation-based SLs from genetic screenings uses a very similar framework to the proposed method SLIdR. Similarly, DAISY and one of the steps in ISLE also uses a somewhat similar framework to SLIdR. Hence a comprehensive comparison of the new method with some of the previously published approaches is missing and should be carried out. Surprisingly, the authors have not even cited some of the previous important methods (like MiSL and ISLE) even though these methods address the same problem and have some similarity to the proposed technique. (One such comparison is suggested in the next point).

2. Objective measurement of sensitivity and specificity is missing from this work, where most of the biological support is hand-picked from the literature and is not done in a systematic and convincing manner. We strongly recommend using publicly available double-knockout screening datasets which could be considered as a gold standard and compute a prediction performance of SLIdR. Further using this framework, a comparison of performance with some of the previous techniques (like MiSL, ISLE, DAISY) should also be performed.

Some examples of gold standard datasets include:

SynLethDB: <http://histone.sce.ntu.edu.sg/SynLethDB/about.php>

Clinically available SLs:

<https://www.sciencedirect.com/science/article/pii/S2405803318302425?via=ihub#tbl0005>

In the work below, 220K SL pairs have been knocked-out in two cell lines for which expression, CNV etc. are characterized. [https://www.cell.com/cell/fulltext/S0092-8674\(18\)30735-9](https://www.cell.com/cell/fulltext/S0092-8674(18)30735-9)

Similarly, in six cell lines known 125 putative SL pairs are knocked out - hence this could possibly a test set. <https://www.nature.com/articles/nbt.4048>

Similarly, there are furthermore experimentally derived SLs in three cell lines each (2600 and 3300 SLs):

Shen et al - <https://www.nature.com/articles/nmeth.4225>

Zhao et al - <https://www.ncbi.nlm.nih.gov/pmc/articles/PMC5449203/>

3. Another straightforward comparison that is missing and could be integrated is CRISPR genetic screenings performed in the exact same cell lines on the exact same genes. Using the same framework, the authors could replace the siRNA essentiality matrix with CRISPR-essentiality matrix to show whether their results are robust across technologies of use. Hits which are robust across both the siRNA and CRISPR screens could be of high-confidence and more robust.

4. Based on the identification of targeting drugs of SL partners identified in Supp Table 1, systematic validation of these hits could also be performed using large-scale drug response profiles as a part of the Project DRIVE used in this work (DepMap).

5. Notably, among pairs identified in this work many genes are SL partners with their own mutation status. This phenomenon is well known and called oncogene addiction and is quite

interesting and the authors may want to give more attention to it and study it deeper. Secondly, some SL pair genes, like NFE2L2 and KEAP1, show a mutually exclusive mutation pattern. This prompts us and suggests that a gene pair, both with an oncogene addiction (SL with their own mutation) and a mutually exclusive mutation profile could be identified as an SL pair by the method, in principle and could confound the results. Adding a step to make sure this is not the case could be one method to resolve this.

6. One of the key hub gene in the overall SL network identified in the work is TP53, where multiple type of mutation occurs including both gain-of-function and loss-of-function. Stratifying the mutation type using prior literature could help improve the signal-to-noise ratio.

Minor comments

1. Description of the framework in the Methods sections needs to be more comprehensive and clearer. For example, the causal inference portion could be better explained.

2. Figure 4d and 4e need to be reordered as per mentioned first in the text.

3. Synthetic lethal interactions computationally identified via this tool seems to be in only one direction and if so, this needs to be clearly stated. That is, SL partners are identified while driver genes are mutated, so the viability comparison is between driver + perturbation knockout vs perturbation gene knockout. There seems to be no comparison of combination viability with the viability of the cell line when the driver gene alone is knocked down.

Reviewer #3 (Remarks to the Author): Expertise in SL and computational biology

In their manuscript entitled "Discovery of synthetic lethal interactions from large-scale pan-cancer perturbation screens" Srivatsa, Montazeri et al. describe a statistical framework (SLIdR) for the identification of synthetic lethal (SL) interactions from public data sets. Specifically, SLIdR is meant to predict SL pairs from small data sets with few false positive predictions. They apply the approach to published RNAi data targeting 7837 genes in 398 cancer cell lines (project DRIVE) and show that their approach can re-identify certain putative SL pairs and predict new pairs. Srivatsa, Montazeri et al. then perform experiments to validate a SL relationship between AXIN1 and URI1 in SNU449 and Huh-7 cells using growth assays and apoptosis measurements. The concept of synthetic lethality has regained traction in recent times now that new technologies enable the study of SL relationships using genetic mutants. Previously, the field has suffered to some extent from RNAi-based studies that identified SL pairs which did not stand the test of time. A method for the robust identification of SL interactions that is broadly applicable to different types of data sets (CRISPR, drugs, RNAi etc.) would therefore represent a valuable resource and important advance in the functional characterization of the cancer genome. This would potentially be of interest to a broad audience which may want to utilize the presented method.

This being said, there are some principal concerns I have with the study that the authors may want to address to convince potential users of the robust nature and superior performance of their approach compared to existing methods/resources.

Major comments:

1. As stated above, there is no shortage in the number of proposed SL relationships from the analysis of large-scale data and several algorithms for the identification of SL gene have been reported. In the current version of the manuscript, I was missing data demonstrating the superiority of the presented approach compared to other SL prediction methods, particularly with regard to the claimed mitigation of false positive reports. It did not become clear to me, what the principle advance of the method described by Srivatsa, Montazeri et al. boils down to. One differentiating factor seems to be that the authors pre-stratify the pan-cancer data based on tissue/cancer type, which seems to yield a much higher number of SL pairs (839) than when the project DRIVE data are analyzed without prior stratification (151). What is the quality of these

additionally identified SL pairs? While I am unable to evaluate the bioinformatics aspects of this work, it would be in the interest of the authors to more clearly present the unique advantages of their method to a broad audience that includes non-bioinformaticians.

2. Out of the substantial number of newly identified SL pairs, the authors select AXIN1/URI1 for experimental validation, despite the fact that their approach also identified SL interactions involving highly relevant cancer genes and druggable factors (examples would be BRAF, PI3K, p53 etc.). The identification of such SL interactions, if proven true, would constitute a major breakthrough in cancer research. The fact that none of these more 'high profile interactions' were chosen for experimental validation somewhat limits the reader's confidence in the data.

3. Related to the previous point, the particular choice of AXIN1/URI1 represents a major problem of the current manuscript in my view. URI1 is a common essential gene (across different cell systems and identified with different approaches, see e.g. depmap portal: Tsherniak et al., Cell, 2017). In line with this, the authors themselves verify the highly essential nature of URI1 with their siRNA experiments: as can be observed in Fig. 4d, siURI1 treated cells are completely blocked in their ability to grow. It is well-known that essential genes are enriched for genetic interactions (e.g. Constanzo et al., Science 2010, 2016) and this represents a major source of undesirable genetic interactions identified in the search of cancer vulnerabilities. The question of essential genes contributing to the identified SL pairs is addressed in similar computational approaches (e.g. SLant, Benstead-Hume et al., PLoS Comput Biol, 2019). To make a convincing case, to my mind, the authors should select new SL pairs that do not contain such 'drop-dead essential genes' for experimental validation. The authors should also equip their method with means to filter against SL pairs involving essential genes.

4. A challenge in the study of genetic interactions is the distinction between synthetic fitness effects from merely additive effects (fitness defect of the SL pair should be greater than the combined fitness defect of the individual mutants). In Figure 4d, the authors show that individual knockdown of AXIN1 or URI1 installs a growth defect in Huh-7 cells. The data do not convince that the combined knockdown in fact creates a synthetic lethal effect. The authors have the respective expertise and this issue should be addressed in the experimental validation of SL gene pairs.

5. Given the well-known complications of RNAi, it is critical that the authors select orthogonal approaches (such as CRISPR or small molecule inhibitors) for the validation of their SL pair predictions based on RNAi data.

6. To support the robustness of their SL prediction approach, ideally the authors should validate a collection of SL interactions experimentally using the aforementioned orthogonal methods. Alternatively, the authors would need to provide some level of mechanistic insight into an identified SL interaction to make a convincing argument.

7. As the scientific community is moving on from RNAi to SL experiments based on genetic mutations, in order for the authors' method to gain traction and be broadly utilized by the scientific community, it would be very valuable if the authors applied their approach also to similar data obtained using CRISPR, such as project Achilles.

Minor comments:

1. The authors cite the genetic interaction of BRCA1/2 and PARP1 in their introduction. Should this relationship be identified by their method?

2. Some of the reported SL pairs include curious combinations, an example would be PI3K and beta-actin (Fig. 2d). How do the authors interpret these findings?

3. The purpose of certain figure panels is unclear, examples include Fig. 1b, Fig. 2a etc. Figure 1 would in general benefit from the inclusion of more data.

4. In my view, the use of siTP53 (Fig. S2) is not suitable to assess the off-target activity of URI1 siRNAs. Additionally, the knockdown efficiency of the employed TP53 siRNA is very modest (Fig.

S2). Again, an orthogonal strategy (e.g. CRISPR-KO or CRISPRi of URI1) or expression of siRNA-resistant URI1 cDNA would be more convincing.

5. There are some additional issues with Fig. S2. Why does siTP53 rescue the growth defect installed by siAXIN1 treatment? Are AXIN1 and TP53 a 'synthetic viable' pair? What is the reason for the observed differences in growth among identically treated cells between Fig. 4d (cell index of siCTRL increases ca. 5-fold over 120h) and Fig. S2 (cell index of siCTRL increases ca. 2-fold over 120h and absolute values are substantially different)?

6. Was FDR-correction applied when multiple t-tests were performed on qPCR/FACS experiments (Fig. 4)?

RESPONSE TO REVIEWERS' COMMENTS

Reviewer #1 (Remarks to the Author): Expertise in SL screens

General comments to the Authors

The authors have developed a method, SLIdR (Synthetic Lethal Identification in R), to identify synthetic lethal (SL) pairs from shRNA genetic screens in cancer cell lines using a novel method and provided an in vitro experimental validation of one predicted SL pair.

The general problem of identifying SL pairs using in both pan-cancer and in a cancer-specific pattern is an important problem. However, in our opinion, this work misses systematic testing of model performance and comparison with available tools, making it very hard to conclude the efficacy of the method and whether it is useful for the community. It also lacks a proper systematic validation of many of the predicted SL pairs. One useful contribution of this work is the specific SL pair (between AXIN1 & URI1) which has been validated in vitro, but for a journal of the level of Nat. Comm. One would expect to see further confirmation in vivo.

Major Comments

1. In the last few years, multiple tools have been published to identify SLs [PMID: ISLE-29959327, DAISY- 25171417, MiSL- 28561042], including a few in Nature Communication. Tools like MiSL (published in Nature Comm.) identifies mutation-based SLs from genetic screenings uses a very similar framework to the proposed method SLIdR. Similarly, DAISY and one of the steps in ISLE also uses a somewhat similar framework to SLIdR. Hence a comprehensive comparison of the new method with some of the previously published approaches is missing and should be carried out. Surprisingly, the authors have not even cited some of the previous important methods (like MiSL and ISLE) even though these methods address the same problem and have some similarity to the proposed technique. (One such comparison is suggested in the next point).

We thank the reviewer for pointing this out. We originally submitted our manuscript as a letter and therefore, had to abridge our introduction and findings to meet the word count requirements. We have since changed the format to the article format in our resubmission and substantially modified the introduction to incorporate the reviewers' suggestions. In particular, we now reference multiple computational tools for identification of the SL pairs in the third

paragraph of Introduction. The corresponding text reads as “*Significant efforts have also been made in developing integrative computational methods ...*”.

Additionally, we performed a comprehensive comparison as per the reviewers’ suggestions, which is discussed in more detail in the next point.

2. Objective measurement of sensitivity and specificity is missing from this work, where most of the biological support is hand-picked from the literature and is not done in a systematic and convincing manner. We strongly recommend using publicly available double-knockout screening datasets which could be considered as a gold standard and compute a prediction performance of SLIdR. Further using this framework, a comparison of performance with some of the previous techniques (like MiSL, ISLE, DAISY) should also be performed.

Some examples of gold standard datasets include:

SynLethDB: <http://histone.sce.ntu.edu.sg/SynLethDB/about.php>

Clinically available SLs:

<https://www.sciencedirect.com/science/article/pii/S2405803318302425?via=ihub#tbl0005>

In the work below, 220K SL pairs have been knocked-out in two cell lines for which expression, CNV etc. are characterized. [https://www.cell.com/cell/fulltext/S0092-8674\(18\)30735-9](https://www.cell.com/cell/fulltext/S0092-8674(18)30735-9)

Similarly, in six cell lines known 125 putative SL pairs are knocked out - hence this could possibly a test set. <https://www.nature.com/articles/nbt.4048>

Similarly, there are furthermore experimentally derived SLs in three cell lines each (2600 and 3300 SLs):

Shen et al - <https://www.nature.com/articles/nmeth.4225>

Zhao et al - <https://www.ncbi.nlm.nih.gov/pmc/articles/PMC5449203/>

Thank you for both comments. We compared SLIdR with the standard Wilcoxon rank sum test, which is adopted by ISLE and DAISY, through a simulation study. The results are explained in the Results section under “SLIdR outperforms conventional tests for SL prediction”. We demonstrate the advantage and superiority of SLIdR by comparing it to the standard Wilcoxon rank sum test for three cancer types (**Supplementary Fig. S2**). The performance plots (as a function of sample size) show that SLIdR consistently reduces the number of false positives even in rare osteosarcomas with only seven cell lines.

In the same section, we also assess and report the performance of SLIdR in identifying gold-standard SL pairs. In particular, we show that with two orders of magnitude fewer predictions than ISLE, SLIdR recovers a comparable fraction of the experimentally identified gold-standard SL interactions reported by (Lee *et al.*, 2018) (**Supplementary Table S4**). This gold-standard list, which was derived from 17 *in vitro* perturbation screens (including a few of

the reviewers' recommendations) by Lee *et al.*, 2018 proved to be a comprehensive list for benchmarking.

3. Another straightforward comparison that is missing and could be integrated is CRISPR genetic screenings performed in the exact same cell lines on the exact same genes. Using the same framework, the authors could replace the siRNA essentiality matrix with CRISPR-essentiality matrix to show whether their results are robust across technologies of use. Hits which are robust across both the siRNA and CRISPR screens could be of high-confidence and more robust.

We thank the reviewer for this suggestion. We applied SLiDR to CRISPR data from Project Achilles in both pan-cancer and cancer type-specific settings. The results are summarized in **Supplementary Table S5** and described in detail in subsections titled "SLiDR on CRISPR dataset" in Methods and "Integration with CRISPR data" in Results. In the pan-cancer analysis of the CRISPR data, we found 104 SL pairs including several pairs from the pan-cancer analysis on DRIVE data. While the overlap between the two screens was significant (hypergeometric p-value $< 10^{-16}$), we observed that several established and strong candidate pairs of the DRIVE analysis were missed in the CRISPR analysis by small margins. One of the primary reasons for this was that of the 373 cell lines from Project DRIVE, ~100 cell lines were missing in the CRISPR dataset. This reduced overlap resulted in a smaller space of possible pairwise interactions and reduced statistical power, particularly in the cancer type-specific analyses. Therefore, to identify the robust hits, we further used Fisher's method to combine the p-values reported by SLiDR for each pair across the two screens and reported the significant ones. In the pan-cancer setting, we identified 162 robust SL pairs across both the screens, out of which we recovered ~62% (91 SL pairs) of the original DRIVE candidate SL pairs, including strong pairs such as, *ACVR2A-WRN*, *RB1-E2F3*, and *RB1-SKP2*, backed by literature evidence.

In addition to identifying robust hits, this analysis enabled us to truly exploit the complementary information across the two screens and proved beneficial in recovering pairs which showed reduced signal in one of the two screens (paragraph 3 of "Integration with CRISPR data" subsection). These results reinforce that RNAi and CRISPR are complementary technologies and together can improve screening and validation of interactions (Mohr *et al.*, 2014; Morgens *et al.*, 2016).

4. Based on the identification of targeting drugs of SL partners identified in Supp Table 1, systematic validation of these hits could also be performed using large-scale drug response profiles as a part of the Project DRIVE used in this work (DepMap).

We thank the reviewer for this suggestion. We systematically validated the predicted pan-cancer and cancer-type specific SL pairs from the Project DRIVE screen on the primary

PRISM drug-response profile (from the DepMap consortium) and elaborated these additional findings in the third paragraph of sub-section “Enrichment of Pan-cancer SL interactions by SLIdR” and third paragraph of “Enrichment of cancer type-specific SL interactions by SLIdR” under Results. The pan-cancer and cancer-type specific hits with support from PRISM drug-response screen have been tabulated in **Supplementary Table S3** and some of them are highlighted in **Figure 2d** and **Figure 3b**.

5. Notably, among pairs identified in this work many genes are SL partners with their own mutation status. This phenomenon is well known and called oncogene addiction and is quite interesting and the authors may want to give more attention to it and study it deeper. Secondly, some SL pair genes, like *NFE2L2* and *KEAP1*, show a mutually exclusive mutation pattern. This prompts us and suggests that a gene pair, both with an oncogene addiction (SL with their own mutation) and a mutually exclusive mutation profile could be identified as an SL pair by the method, in principle and could confound the results. Adding a step to make sure this is not the case could be one method to resolve this.

We thank the reviewer for the constructive criticism. While oncogene addiction is an interesting concept, the primary goal of our method is to identify genetic interactions consisting of druggable targets in cancers driven by undruggable oncogenes. In a parallel study (Montazeri *et al.*, no date) focus on identifying such drivers from perturbation screens.

Regarding the mutually exclusive mutation pattern shown by some SL pairs, we would like to point out that such a pattern has been suggested to be a strong indicator of synthetic lethal interactions. Indeed, loss of one gene in a synthetic lethal gene pair creates a dependency on the other gene, thus synthetic lethal gene pairs are not expected to show simultaneous loss-of-function, and, on the contrary, usually show a mutual exclusive mutation pattern (Muller *et al.*, 2015). This principle is the foundation of large-scale functional genomic screenings (Wappett *et al.*, 2016) and is used in several methods for identifying SL pairs. We therefore decided to include such pairs of genes since we believe this makes the method more informative. Especially with regard to the *NFE2L2* and *KEAP1* gene pair, it is well-known that *KEAP1* is a negative regulator of Nrf2 (*NFE2L2*) transcription factor. Loss-of-function mutations in the *KEAP1* gene lead to Nrf2 stress pathway activation and addiction towards this pathway. From a biological point of view, it therefore makes sense that *NFE2L2* and *KEAP1* appear to be a synthetic lethal pair. This can also be connected to the first point addressed by the reviewer regarding oncogene addiction. While classical oncogene addiction refers to a situation where cancer cells are addicted to a specific activating mutation in a given oncogene, synthetic lethality can derive from a similar situation where inactivation of a given tumor suppressor gene can activate an oncogenic pathway. We therefore believe that adding a further step might not be beneficial to the SLIdR method.

6. One of the key hub gene in the overall SL network identified in the work is TP53, where multiple type of mutations occur including both gain-of-function and loss-of-function. Stratifying the mutation type using prior literature could help improve the signal-to-noise ratio.

We thank the reviewer for raising this point. As suggested, we stratified cell lines based on the TP53 mutation type using established literature in the pan-cancer analysis. However, the results remained unchanged post-stratification since the majority of the original 373 cell lines harbored loss-of-function mutations and only a very small fraction (3.7%) exhibited gain-of-function mutations. Consequently, the post-stratification results recovered all the 151 SL pairs as prior to stratification. Since the results did not change, we did not include it in the updated manuscript.

Minor comments

1. Description of the framework in the Methods sections needs to be more comprehensive and clearer. For example, the causal inference portion could be better explained.

To improve the comprehensibility of the framework, we elaborated on the SLIdR algorithm in the subsection titled “The SLIdR workflow” under Results. We also updated the second and third paragraph in the “Causal inference” subsection under Methods for clarity.

2. Figure 4d and 4e need to be reordered as per mentioned first in the text.

Thank you for pointing this out! The figures and their labels have since changed in the revised manuscript.

3. Synthetic lethal interactions computationally identified via this tool seems to be in only one direction and if so, this needs to be clearly stated. That is, SL partners are identified while driver genes are mutated, so the viability comparison is between driver + perturbation knockout vs perturbation gene knockout. There seems to be no comparison of combination viability with the viability of the cell line when the driver gene alone is knocked down.

We agree with the reviewer that it would be appropriate to check the synthetic lethality in both directions. However, since many driver gene knockdowns are missing in the perturbation screening data, this is not feasible for all pairs and was therefore not included in our framework. We have explicitly stated this at the end of the subsection titled “SLIdR algorithm” under Methods, in the revised manuscript. We write *“It is important to note that since knockdown data of several driver genes is unavailable in the Project DRIVE screen, we tested for SL*

interactions only in one direction and were unable to test the effect of viability of the cell lines when the driver genes alone are knocked down."

Reviewer #3 (Remarks to the Author): Expertise in SL and computational biology

In their manuscript entitled "Discovery of synthetic lethal interactions from large-scale pan-cancer perturbation screens" Srivatsa, Montazeri et al. describe a statistical framework (SLIdR) for the identification of synthetic lethal (SL) interactions from public data sets. Specifically, SLIdR is meant to predict SL pairs from small data sets with few false positive predictions. They apply the approach to published RNAi data targeting 7837 genes in 398 cancer cell lines (project DRIVE) and show that their approach can re-identify certain putative SL pairs and predict new pairs. Srivatsa, Montazeri et al. then perform experiments to validate a SL relationship between AXIN1 and URI1 in SNU449 and Huh-7 cells using growth assays and apoptosis measurements. The concept of synthetic lethality has regained traction in recent times now that new technologies enable the study of SL relationships using genetic mutants. Previously, the field has suffered to some extent from RNAi-based studies that identified SL pairs which did not stand the test of time. A method for the robust identification of SL interactions that is broadly applicable to different types of data sets (CRISPR, drugs, RNAi etc.) would therefore represent a valuable resource and important advance in the functional characterization of the cancer genome. This would potentially be of interest to a broad audience which may want to utilize the presented method.

This being said, there are some principal concerns I have with the study that the authors may want to address to convince potential users of the robust nature and superior performance of their approach compared to existing methods/resources.

Major comments:

1. As stated above, there is no shortage in the number of proposed SL relationships from the analysis of large-scale data and several algorithms for the identification of SL gene have been reported. In the current version of the manuscript, I was missing data demonstrating the superiority of the presented approach compared to other SL prediction methods, particularly with regard to the claimed mitigation of false positive reports. It did not become clear to me, what the principle advance of the method described by Srivatsa, Montazeri et al. boils down to. One differentiating factor seems to be that the authors pre-stratify the pan-cancer data based on tissue/cancer type, which seems to yield a much higher number of SL pairs (839) than when the project DRIVE data are analyzed without prior stratification (151). What is the quality of

these additionally identified SL pairs? While I am unable to evaluate the bioinformatics aspects of this work, it would be in the interest of the authors to more clearly present the unique advantages of their method to a broad audience that includes non-bioinformaticians.

We thank the reviewer for this constructive criticism and apologize for the lack of clarity. As mentioned earlier, we originally submitted our manuscript as a letter and therefore, had to make it concise to meet the word count requirements. We have revised the manuscript with an emphasis on the advantages of our method over others.

Methods for identifying SL interactions primarily rely on large sample sizes and multi-omics data and thus are not suitable for rare cancer types. Furthermore, they regard the shRNA screening data to have low statistical power and therefore, use genetic perturbation screens merely to refine the candidate list of SL pairs derived from analyzing multi-omics data (Jerby-Arnon *et al.*, 2014). However, we show that such screens can be highly informative in identifying SL pairs directly. Through a simulation study, we demonstrate in the revised manuscript the advantage and superiority of SLIdR for three cancer types (**Supplementary Fig. S2**). The performance plots (as a function of sample size) show that SLIdR is consistently reducing the number of false positives even in rare osteosarcomas with only seven cell lines. Further, we show that with two orders of magnitude fewer predictions than the competing SL prediction method ISLE, SLIdR recovers a comparable fraction of the experimentally identified gold-standard SL interactions reported by (Lee *et al.*, 2018) (**Supplementary Table S4**). These results are explained in detail in sub-section “SLIdR outperforms conventional tests for SL prediction”.

Regarding the differences between pan-cancer and cancer type-specific analyses, both are important as they provide complementary information, and SLIdR can successfully identify well-established and novel targets in both these settings, as demonstrated. Pan-cancer analysis is desirable for identifying SL partners for mutated driver genes shared across multiple cancer types, while cancer-type specific analysis identifies SL partners for specific or rare mutated driver genes. We have modified the manuscript and elaborated further on these results in the revised manuscript in sub-sections “Enrichment of Pan-cancer SL interactions by SLIdR” and “Enrichment of cancer type-specific SL interactions by SLIdR”.

2. Out of the substantial number of newly identified SL pairs, the authors select AXIN1/URI1 for experimental validation, despite the fact that their approach also identified SL interactions involving highly relevant cancer genes and druggable factors (examples would be BRAF, PI3K, p53 etc.). The identification of such SL interactions, if proven true, would constitute a major break-through in cancer research. The fact that none of these more ‘high profile interactions’ were chosen for experimental validation somewhat limits the reader’s confidence in the data.

We thank the reviewer for this insightful comment. We have performed additional experimental validation of novel SL pairs in HCC, a cancer type with few targeted therapies available (Llovet *et al.*, 2018). In particular, we validated AXIN1-URI1, SLIdR's top SL hit in HCC, in the original manuscript. In addition, to address the reviewer's comment, we successfully validated ARID1A-TEAD1, in the revised manuscript. The results are explained in **Figures 4 and 5**, and **Supplementary Figure S3**. In detail, we proved that inhibition of TEAD1, using siRNA or the small molecule inhibitor verteporfin, induces cell death in ARID1A-mutant HCC cells in a dose- and time-dependent manner (**Figure 4**). Silencing of ARID1A makes ARID1A-wild-type HCC cells susceptible to TEAD1 inhibition via siRNA or verteporfin, both *in vitro* and *in vivo* (**Figure 5**, **Supplementary Figure S3**). As written in the revised manuscript in the Discussion: “Taken together our results show that ARID1A and TEAD1 are synthetic lethal interactors in HCC and strongly indicate that this relationship is dependent on the regulatory function of TEAD1 in the Hippo signalling pathway. Indeed, our results are in accordance with the recent work of Chang *et al.*⁵³. Given the widespread role of the SWI/SNF complex⁵⁴, the frequency of ARID1A inactivation in several malignancies other than HCC⁵⁴ and availability of TEAD-YAP inhibitors⁴⁶, the identification of the ARID1A-TEAD1 synthetic lethal pair provides an example of how SLIdR can help improve cancer therapy.”

In addition, based on the predictions by SLIdR, Bianco *et al.* validated GATA3-MDM2 as an SL pair in ER-positive breast cancer in a separate extensive *in vitro* and *in vivo* study (Bianco *et al.*, no date). With MDM2 inhibitors widely available (Konopleva *et al.*, 2020), this discovery further emphasizes the ability of SLIdR to predict new cancer-specific druggable targets. While validating other synthetic lethal interactions might have broader applications in cancer research, further experimental validation is beyond the scope of the present study.

3. Related to the previous point, the particular choice of AXIN1/URI1 represents a major problem of the current manuscript in my view. URI1 is a common essential gene (across different cell systems and identified with different approaches, see e.g. depmap portal: Tsherniak *et al.*, Cell, 2017). In line with this, the authors themselves verify the highly essential nature of URI1 with their siRNA experiments: as can be observed in Fig. 4d, siURI1 treated cells are completely blocked in their ability to grow. It is well-known that essential genes are enriched for genetic interactions (e.g. Constanzo *et al.*, Science 2010, 2016) and this represents a major source of undesirable genetic interactions identified in the search of cancer vulnerabilities. The question of essential genes contributing to the identified SL pairs is addressed in similar computational approaches (e.g. SLant, Benstead-Hume *et al.*, PLoS Comput Biol, 2019). To make a convincing case, to my mind, the authors should select new SL pairs that do not contain such ‘drop-dead essential genes’ for experimental validation. The authors should also equip their method with means to filter against SL pairs involving essential genes.

We thank the reviewer for the constructive criticism. The literature on *URI1* has mixed comments on its function. While, as mentioned by the reviewer, many independent lines of evidence point at *URI1* as an essential gene, several others clearly demonstrated that it acts as an oncogene in different tumor types (Theurillat *et al.*, 2011), and specifically HCC (Tummala *et al.*, 2014; Zhang *et al.*, 2015).

As the reviewer noted, we indeed showed that silencing of *URI1* alone significantly affected cell proliferation in Huh-7 cells (**Supplementary Fig.S4e**), which is also expected given the oncogenic role of *URI1* in HCC. However, we additionally showed that while *URI1* silencing alone does not impact cell death, dual silencing of *URI1* and *AXIN1* significantly induced apoptosis (**Supplementary Fig.S4f**). Therefore, our data suggests that *URI1*-silencing affects cell proliferation, but *URI1* is not an essential gene in HCC, given that cells are still fully alive upon gene knock-down. On the contrary, the apoptotic phenotype induced by dual silencing of *URI1* and *AXIN1* suggests a synthetic lethal interaction.

Considering the controversy on *URI1* function, we accepted the suggestion of the reviewer and decided to validate a second SL pair. Specifically, we validated the synthetic lethal interaction between *ARID1A* and *TEAD1* in HCC, using both a siRNA approach and the use of TEAD1-YAP inhibitor (verteporfin). We validated our results both *in vitro* and *in vivo* (**Figure 4, Figure 5, Supplementary Fig. S3**).

Finally, the method indeed filters essential genes in the pre-processing step (under **Viability data from perturbation screens in Methods**). As recommended by (McDonald *et al.*, 2017), genes with RSA value ≤ -3 in more than 50% of cancer cell lines were considered as essential genes and filtered from the viability data.

4. A challenge in the study of genetic interactions is the distinction between synthetic fitness effects from merely additive effects (fitness defect of the SL pair should be greater than the combined fitness defect of the individual mutants). In Figure 4d, the authors show that individual knockdown of *AXIN1* or *URI1* installs a growth defect in Huh-7 cells. The data do not convince that the combined knockdown in fact creates a synthetic lethal effect. The authors have the respective expertise and this issue should be addressed in the experimental validation of SL gene pairs.

We agree with the reviewer that the proliferation assay may not be convincing of the synthetic lethal interaction. Therefore, we performed an apoptotic assay and showed that cells die only upon dual silencing of *URI1* and *AXIN1*, while no difference in the fraction of live or apoptotic

cells is detected upon single gene knock-down (**Supplementary Fig.S4f**). These data support the prediction of *URI1-AXIN1* as a synthetic lethal pair in HCC.

5. Given the well-known complications of RNAi, it is critical that the authors select orthogonal approaches (such as CRISPR or small molecule inhibitors) for the validation of their SL pair predictions based on RNAi data.

We thank the reviewer for the observation. In the revised manuscript, we used an orthogonal method to validate the *ARID1A-TEAD1* SL pair. Specifically we have used verteporfin, a small molecule inhibitor which blocks the interaction between the co-activator *YAP* and *TEAD1*, thus inhibiting *TEAD1* transcription activity (Liu-Chittenden *et al.*, 2012; Feng *et al.*, 2016). Using this method we showed that inhibition of TEAD transcription activity, specifically the one related to the Hippo signalling pathway, is lethal in the context of ARID1A-mutant or ARID1A-silenced cells (**Figure 4 and 5, Supplementary Figure S3**). In particular, we showed that treatment with verteporfin affects tumor growth of ARID1A-silenced cells *in vitro* and *in vivo*, while having little to no effect on control cells. A recent discovery indicated that *ARID1A* plays a role in regulating the interaction between YAP and TEAD factors (Chang *et al.*, 2018), thus supporting our conclusion.

Beyond the experimental validation, in the revised manuscript, we broadly validated the predicted pan-cancer and cancer-type specific SL pairs from the Project DRIVE screen on the primary PRISM drug-response profile (from the DepMap consortium). The pan-cancer and cancer-type specific hits with support from PRISM drug-response screen have been tabulated in **Supplementary Table S3** and some of them are highlighted in **Figure 2d** and **Figure 3b**. Furthermore, we also applied SLiDR to CRISPR data from the Project Achilles in both pan-cancer and cancer type-specific settings. The results are summarized in **Supplementary Table S5** and described in detail in subsections titled “SLiDR on CRISPR dataset” in Methods and “Integration with CRISPR data” in Results. These comparisons and validations have been addressed in detail under reviewer 1’s comments 3 and 4.

6. To support the robustness of their SL prediction approach, ideally the authors should validate a collection of SL interactions experimentally using the aforementioned orthogonal methods. Alternatively, the authors would need to provide some level of mechanistic insight into an identified SL interaction to make a convincing argument.

To support the robustness of SLiDR predictions, we validated two independent synthetic lethal pairs in HCC. For the validation of *ARID1A-TEAD1*, we specifically used an orthogonal method (the small molecule inhibitor verteporfin). Our data also suggest that this synthetic lethality relies on the Hippo signalling pathway, specifically on the binding of TEAD and YAP. Additionally,

Bianco *et al.* (Bianco *et al.*, no date) recently validated the *GATA3-MDM2* predicted pair in ER-positive breast cancer using siRNA and the small molecule inhibitor Idasanutlin, in a comprehensive work including in vitro, in vivo, and ex-vivo (patient-derived organoids) approaches. They have additionally provided some mechanistic insight into the SL pair and showed that it is at least partially due to activation of the mTOR signalling pathway. Further, we found the same pair when using the CRISPR screening data. Such extensive validation requires time and resources and we believe that it constitutes an independent research line, such as the one performed by Bianco *et al.* With in total three independent synthetic lethal pairs validated, we believe to have provided strong proof of principle of the predictive power of SLIdR. The validation of a collection of SL would be beyond the scope of the presented manuscript.

7. As the scientific community is moving on from RNAi to SL experiments based on genetic mutations, in order for the authors' method to gain traction and be broadly utilized by the scientific community, it would be very valuable if the authors applied their approach also to similar data obtained using CRISPR, such as project Achilles.

We thank the reviewer for this suggestion. This point has been addressed in detail under reviewer 1's comment 3. In summary, we applied SLIdR to CRISPR data from the Project Achilles in both pan-cancer and cancer type-specific settings. The results are summarized in **Supplementary Table S5** and described in detail in subsections titled "SLIdR on CRISPR dataset" in Methods and "Integration with CRISPR data" in Results.

Minor comments:

1. The authors cite the genetic interaction of BRCA1/2 and PARP1 in their introduction. Should this relationship be identified by their method?

Thank you for raising this question. This pair is not in the space of possible pairwise interactions in our dataset and therefore could not be identified. Specifically, we did not have BRCA1/2 in our mutation profiles and therefore could not assess this pair.

2. Some of the reported SL pairs include curious combinations, an example would be PI3K and beta-actin (Fig. 2d). How do the authors interpret these findings?

Regarding the PI3K and beta-actin interaction, we agree with the reviewer that the interaction might appear unexpected at first glance. However, we speculate that the interaction might arise from the interconnection between the PI3K-Akt-mTOR pathway and signalling mediated by the actin cytoskeleton. Synthetic lethal pairs typically involve genes controlling the same downstream pathway. In this specific case, the PI3K-mTOR signalling pathway is a well-studied

regulator of the organization of the actin cytoskeleton (Ho *et al.*, 2011; Marshansky, 2016; Jing *et al.*, 2020). Supporting these data and the SL interaction between PI3K and beta-actin, inhibition of PI3K has been shown to induce apoptosis in colorectal cancer cells, even in the presence of activating mutant PIK3CA, by inducing a Rac-independent actin rearrangement (Mallucci *et al.*, 2012). However, we can only speculate on our finding, and as for every other pair, systematic experimental validation will be required.

Like with any prediction based on statistical models, of course, some of the hits reported by SLIdR are expected to be false positives due to the procedure itself and various (known and unknown) confounders. We have explicitly discussed this issue in the Discussion section under limitations.

3. The purpose of certain figure panels is unclear, examples include Fig. 1b, Fig. 2a etc. Figure 1 would in general benefit from the inclusion of more data.

A key advantage of the SLIdR is its ability to predict SL pairs with small sample size. Figure 1b was added to show the distribution of sample sizes in the DRIVE data across various cancer types. Since many cancers have small sample sizes, the figure shows the necessity of developing a computational tool with sufficient statistical power on small sample size data.

Figure 2a indicates the frequencies of 84 mutated driver genes across different cancer types. We have included this figure as it shows frequency distribution as well as diversity of mutation data. In addition, the obtained SL p-values from the SLIdR, to some extent, depend on the number of mutated cell lines for the corresponding driver gene; hence this figure gives some useful information for better interpretation of the SLIdR SL hits. The corresponding captions were slightly modified for further clarification.

4. In my view, the use of siTP53 (Fig. S2) is not suitable to assess the off-target activity of URI1 siRNAs. Additionally, the knockdown efficiency of the employed TP53 siRNA is very modest (Fig. S2). Again, an orthogonal strategy (e.g. CRISPR-KO or CRISPRi of URI1) or expression of siRNA-resistant URI1 cDNA would be more convincing.

We apologize for the lack of clarity. We employed the *AXIN1-TP53* as a non-SL pair (negative control), in order to prove the specificity of the SLIdR prediction, rather than assessing the off-target activity of *URI1* siRNAs. To avoid off-target activity of *URI1* siRNAs, we used the ON-TARGET plus SMARTpool siRNAs against human *URI1* and ON-TARGET plus SMARTpool non-targeting control from Dharmacon. ON-TARGET plus SMARTpools comprise 4 individual siRNAs and are known to reduce off-targets by up to 90% compared to other siRNAs. The specificity of the SMARTpools is achieved by two strategies: “pooling” and specific pattern

modifications. The Dharmacon research group was the first to experimentally demonstrate the key role of the seed region in mediating off-targets (Birmingham *et al.*, 2006; Anderson *et al.*, 2008). These principles were subsequently applied by Dharmacon researchers to design the ON-TARGET SMARTpools and to improve specificity (Jackson *et al.*, 2006). Additionally, ON-TARGET*plus* can be used at a very low concentration (from 5 to 25 nM) which also helps reduce off-targets effects.

We agree with the reviewer's suggestion and have used a small molecule inhibitor, in addition to siRNAs, for experimental validation of *ARID1A-TEAD1* in the revised manuscript (**Fig.4 and Fig.5** in the revised manuscript). Moreover, using different concentrations of *TEAD1* siRNAs, we showed that the phenotype induced by the knock-down of *TEAD1* in *ARID1A*-mutant cells was dose and time-dependent, thus indicating a specific on-target effect (**Fig. 4**).

5. There are some additional issues with Fig. S2. Why does siTP53 rescue the growth defect installed by siAXIN1 treatment? Are AXIN1 and TP53 a 'synthetic viable' pair? What is the reason for the observed differences in growth among identically treated cells between Fig. 4d (cell index of siCTRL increases ca. 5-fold over 120h) and Fig. S2 (cell index of siCTRL increases ca. 2-fold over 120h and absolute values are substantially different)?

While the differences in cell proliferation between *TP53*-silencing alone and dual silencing are only minimal and significant only 120 hours post seeding, we cannot exclude the hypothesis of *AXIN1-TP53* being a "synthetic viable pair". However, SLiDR has not been designed to predict such pairs. We would like to clarify that this pair was selected as a non-SL pair (negative control), to demonstrate the specificity of the method.

Regarding the observed differences in growth among identically treated cells, these are due to the fact that treated cells are seeded at different times and conditions (batch effects; e.g. different cell passages, different cell clones).

6. Was FDR-correction applied when multiple t-tests were performed on qPCR/FACS experiments (Fig. 4)?

Thank you for this comment! Yes, we have used the two-stage step-up method of Benjamini, Krieger and Yekutieli (Benjamini, Krieger and Yekutieli, 2006) for FDR correction (FDR significance level: 1%). To clarify this, we added a new paragraph in the method section "Quantification and statistical analysis of experimental validation".

References

- Anderson, E. M. *et al.* (2008) 'Experimental validation of the importance of seed complement frequency to siRNA specificity', *RNA*, 14(5), pp. 853–861.
- Benjamini, Y., Krieger, A. M. and Yekutieli, D. (2006) 'Adaptive linear step-up procedures that control the false discovery rate', *Biometrika*, pp. 491–507. doi: 10.1093/biomet/93.3.491.
- Bianco, G. *et al.* (no date) 'GATA3 and MDM2 are synthetic lethal in estrogen receptor-positive breast cancers'. doi: 10.1101/2020.05.18.101998.
- Birmingham, A. *et al.* (2006) '3' UTR seed matches, but not overall identity, are associated with RNAi off-targets', *Nature methods*, 3(3), pp. 199–204.
- Chang, L. *et al.* (2018) 'The SWI/SNF complex is a mechanoregulated inhibitor of YAP and TAZ', *Nature*, 563(7730), pp. 265–269.
- Feng, J. *et al.* (2016) 'Verteporfin, a suppressor of YAP-TEAD complex, presents promising antitumor properties on ovarian cancer', *OncoTargets and therapy*, 9, pp. 5371–5381.
- Ho, Y.-P. *et al.* (2011) 'β-Actin is a downstream effector of the PI3K/AKT signaling pathway in myeloma cells', *Molecular and cellular biochemistry*, 348(1-2), pp. 129–139.
- Jackson, A. L. *et al.* (2006) 'Position-specific chemical modification of siRNAs reduces "off-target" transcript silencing', *RNA*, 12(7), pp. 1197–1205.
- Jerby-Arnon, L. *et al.* (2014) 'Predicting cancer-specific vulnerability via data-driven detection of synthetic lethality', *Cell*, 158(5), pp. 1199–1209.
- Jing, Y. *et al.* (2020) 'STING couples with PI3K to regulate actin reorganization during BCR activation', *Science advances*, 6(17), p. eaax9455.
- Konopleva, M. *et al.* (2020) 'MDM2 inhibition: an important step forward in cancer therapy', *Leukemia*, 34(11), pp. 2858–2874.
- Lee, J. S. *et al.* (2018) 'Harnessing synthetic lethality to predict the response to cancer treatment', *Nature communications*, 9(1), p. 2546.
- Liu-Chittenden, Y. *et al.* (2012) 'Genetic and pharmacological disruption of the TEAD-YAP complex suppresses the oncogenic activity of YAP', *Genes & development*, 26(12), pp. 1300–1305.
- Llovet, J. M. *et al.* (2018) 'Molecular therapies and precision medicine for hepatocellular carcinoma', *Nature Reviews Clinical Oncology*, pp. 599–616. doi: 10.1038/s41571-018-0073-4.
- Mallucci, L. *et al.* (2012) 'Killing of Kras-mutant colon cancer cells via Rac-independent actin remodeling by the βGBP cytokine, a physiological PI3K inhibitor therapeutically effective in vivo',

Molecular cancer therapeutics, 11(9), pp. 1884–1893.

Marshansky, V. (2016) 'Faculty Opinions recommendation of Phosphoinositide 3-Kinase Regulates Glycolysis through Mobilization of Aldolase from the Actin Cytoskeleton', *Faculty Opinions – Post-Publication Peer Review of the Biomedical Literature*. doi: 10.3410/f.726107259.793518086.

McDonald, E. R., 3rd *et al.* (2017) 'Project DRIVE: A Compendium of Cancer Dependencies and Synthetic Lethal Relationships Uncovered by Large-Scale, Deep RNAi Screening', *Cell*, 170(3), pp. 577–592.e10.

Mohr, S. E. *et al.* (2014) 'RNAi screening comes of age: improved techniques and complementary approaches', *Nature reviews. Molecular cell biology*, 15(9), pp. 591–600.

Montazeri, H. *et al.* (no date) 'Systematic Identification of Novel Cancer Genes through Analysis of Deep shRNA Perturbation Screens'. doi: 10.1101/807248.

Morgens, D. W. *et al.* (2016) 'Systematic comparison of CRISPR/Cas9 and RNAi screens for essential genes', *Nature biotechnology*, 34(6), pp. 634–636.

Muller, F. *et al.* (2015) 'Corrigendum: Passenger deletions generate therapeutic vulnerabilities in cancer', *Nature*, 525(7568), p. 278.

Theurillat, J.-P. *et al.* (2011) 'URI is an oncogene amplified in ovarian cancer cells and is required for their survival', *Cancer cell*, 19(3), pp. 317–332.

Tummala, K. S. *et al.* (2014) 'Inhibition of de novo NAD(+) synthesis by oncogenic URI causes liver tumorigenesis through DNA damage', *Cancer cell*, 26(6), pp. 826–839.

Wappett, M. *et al.* (2016) 'Multi-omic measurement of mutually exclusive loss-of-function enriches for candidate synthetic lethal gene pairs', *BMC genomics*, 17, p. 65.

Zhang, J. *et al.* (2015) 'RMP promotes venous metastases of hepatocellular carcinoma through promoting IL-6 transcription', *Oncogene*, 34(12), pp. 1575–1583.

REVIEWER COMMENTS

Reviewer #1

General comments

We carefully read the author's response to our initial review regarding their paper titled "Discovery of synthetic lethal interactions from large-scale pan-cancer perturbation screens". While we appreciate their efforts to address our comments, we still have some major concerns. In general, the "response to reviewer document" needs to be significantly improved. Specifically, the details regarding methods and results in a subset of responses are missing or wrongly referenced hindering our ability to judge the quality of the response. A comprehensive and well-structured response may help to overcome this. Based on this response, a major part of what was done was not clear to us. The manuscript is also not well-written.

A few main concerns regarding validations are either not satisfactorily addressed or our ability to judge is hindered due to the unclear presentation of the results. Though we acknowledge and commend the authors on their thorough effort addressing these comments partly and improve the overall manuscript. We recommend the authors to provide a revised version of the manuscript addressing our comments below before further consideration.

Specific comments

- 1) The simulated studies in point 2 (response to reviewer document which is in response to our 'major comments 1 and 2'), complement and supports the methodology quite well. Though we find the description of this section quite poor in both the response document as well as in the manuscript. It was hard to follow and understand what the authors did. Specifically:
 - a. We would suggest the authors provide a high-level description of the method (please follow this in general for other analysis as well) in the results section to guide and help the reader understand.
 - b. The motivation behind this analysis was not clear.
 - c. How the control test was decided and What do the authors mean by saying they used the standard Wilcoxon test initially?

The second portion of this response is unsatisfactorily and lacks the needed depth. This step is critical to comprehensively demonstrate a validation of the methodology. Specifically, we provided a comprehensive list of gold-standard sets for the authors consideration, but they ignored mostly it. We would like to note that Horlbeck et al., 2018, a screen among the listed ones we provided ([https://www.cell.com/cell/fulltext/S0092-8674\(18\)30735-9](https://www.cell.com/cell/fulltext/S0092-8674(18)30735-9)) contains a total of 220K pairs experimentally tested and could provide quite a large and significant set to test their method. They could have been used to generate positive and negative SL sets (gold-standard) and then used for validation, using prediction measures like ROC-AUC or Area under precision recall or some other performance measure. We must note that they are not specifically focused on driver genes and thus need to be carefully done.

While we appreciate the analysis done on the gold-standard SL interactions identified by Lee et al., 2018; there are some major caveats in choosing this set.

- (i) Supp. Table S4 suggests that ISLE provides superior hypergeometric test results compared to SLIdR ($P = 1.44E-28$ for ISLE vs $2.76E-10$ for SLIdR). Is this an error? Also, it is surprising that ISLE identifies over 16 million SL pairs, when the original publication identifies only a few thousand? Please provide more details on how the ISLE was run?
- (ii) The initial set chosen comprises siRNA screens performed in the early decade. The quality of these screens were shown to be very poor due to the off-target effect of siRNA and thus not appropriate for gold-standard set (PMCID: 10.1186/s11658-019-0196-3 , <https://doi.org/10.1038/s41598-017-18551-z> , <https://doi.org/10.1371/journal.pbio.2003213>). Considering this, we think Lee et al.'s step to consider this SL set as an initial pool is okay, though choosing this set as gold-standard is not justified.

We suggest the authors to please run a validation step considering some of the following screens. Please find below a list of recent combinatorial CRISPR screening studies:

- Gonatopoulos-Pournatzis, T., Aregger, M., Brown, K.R. et al. Genetic interaction mapping and exon-resolution functional genomics with a hybrid Cas9–Cas12a platform. *Nat Biotechnol* 38, 638–648 (2020)
- Aregger, M., Lawson, K.A., Billmann, M. et al. Systematic mapping of genetic interactions for de novo fatty acid synthesis identifies C12orf49 as a regulator of lipid metabolism. *Nat Metab* 2, 499–513 (2020).
- Thompson, N.A., Ranzani, M., van der Weyden, L. et al. Combinatorial CRISPR screen identifies fitness effects of gene paralogues. *Nat Commun* 12, 1302 (2021).
- DeWeirdt, P.C., Sanson, K.R., Sangree, A.K. et al. Optimization of AsCas12a for combinatorial genetic screens in human cells. *Nat Biotechnol* 39, 94–104 (2021).
- Dede, M., McLaughlin, M., Kim, E. et al. Multiplex enCas12a screens detect functional buffering among paralogs otherwise masked in monogenic Cas9 knockout screens. *Genome Biol* 21, 262 (2020).
- Gier, R.A., Budinich, K.A., Evitt, N.H. et al. High-performance CRISPR-Cas12a genome editing for combinatorial genetic screening. *Nat Commun* 11, 3455 (2020).
- Replogle, J.M., Norman, T.M., Xu, A. et al. Combinatorial single-cell CRISPR screens by direct guide RNA capture and targeted sequencing. *Nat Biotechnol* 38, 954–961 (2020).
- DeWeirdt, P.C., Sangree, A.K., Hanna, R.E. et al. Genetic screens in isogenic mammalian cell lines without single cell cloning. *Nat Commun* 11, 752 (2020).

- Liu, J. et al. Pooled library screening with multiplexed Cpf1 library. *Nat Commun* 10, 3144 (2019).
- Zhao, Y., Tyrishkin, K., Sjaarda, C. et al. A one-step tRNA-CRISPR system for genome-wide genetic interaction mapping in mammalian cells. *Sci Rep* 9, 14499 (2019).
- Norman, Horlbeck, Replodge, Ge, Xu, Jost, Gilbert, Weissman, Exploring genetic interaction manifolds constructed from rich single-cell phenotypes, *Science* 786-793 (2019).
- Boettcher, M. et al. Dual gene activation and knockout screen reveals directional dependencies in genetic networks. *Nat. Biotechnol.*36, 170–178 (2018).
- Zhao, Badur, Luebeck, Magaña, Birmingham, Sasik, Ahn, Ideker, Metallo, Mali, Combinatorial CRISPR-Cas9 Metabolic Screens Reveal Critical Redox Control Points Dependent on the KEAP1-NRF2 Regulatory Axis, *Molecular Cell*, Volume 69, Issue 4, 699-708.e7 (2018).
- Horlbeck, M. A. et al. Mapping the genetic landscape of human cells. *Cell*174, 953–967.e22 (2018).
- Han, K. et al. Synergistic drug combinations for cancer identified in a CRISPR screen for pairwise genetic interactions. *Nat. Biotechnol.*35, 463–474 (2017).
- Najm, F. J. et al. Orthologous CRISPR–Cas9 enzymes for combinatorial genetic screens. *Nat. Biotechnol.*36, 179–189 (2017).
- Shen, J. P. et al. Combinatorial CRISPR–Cas9 screens for de novo mapping of genetic interactions. *Nat. Methods*14, 573–576 (2017).
- Wong, A. S. L. et al. Multiplexed barcoded CRISPR-Cas9 screening enabled by CombiGEM. *Proc. Natl Acad. Sci. USA*113, 2544–2549 (2016).

- 2) Regarding the authors' response to our original "major comment 4": The drug response prediction results are poorly presented hindering our ability to judge the results and are not convincing. Specifically
- a. Multiple hypothesis correction (FDR) needs to be done, otherwise these p-values could be false positives.
 - b. Appropriate control experiments using random SL pairs could also be carried out.
 - c. Fig. 2d is referenced in the results of pan-cancer testing. We did not find any pan-cancer results in figure 2d. Please provide this separately.
 - d. In addition, Fig. 2d needs improvement. Specifically, most of the p-values are blue colored and based on the current legend it is not clear whether they are significant or not. Fig. 3b does not indicate drug response validation. We did not find Fig. 3b to be informative, a simpler figure showing the number of drugs which pass the FDR threshold can be shown for various drugs or volcano plots showing controls and SLs together providing significance and effect size together. Drug response prediction can be shown either in terms of p-values (after FDR correction), or ROC-AUCs; based on whatever the authors deem fit.

In general, we understand that drug response is a hard question and therefore this is not critical. Thus, it is okay even if SLIdR does not work well in predicting drug response, given the heterogeneity and noise of screens or provides a very low coverage after FDR is correct, it needs to be explicitly mentioned.

3) We are fine with the rest of the responses.

We thank and commend the authors for their effort in addressing these comments and improving their manuscript.

Reviewer #3 (Remarks to the Author):

In their revised manuscript, Srivatsa et al. have addressed several of my earlier concerns, which in my eyes has strengthened their work. Most importantly, they have elaborated on what distinguishes their method from other methods of SL pair prediction and have included additional validation. They have addressed my concern about AXIN1/URI1 by moving these data to the supplement and introduced a new SL pair ARID1A/TEAD1 for the validation experiments and in their rebuttal state that investigation of the 'high profile' SL pairs identified by their approach would be beyond the scope of the study. Overall, the authors are to be complemented for their additional efforts and the new data help to make a stronger case for their method. However, the introduction of these new data also comes with some new concerns that should be addressed prior to publication.

Major points:

1. The new figure 4 is entirely dedicated to studying the effect of TEAD1 inhibition in an ARID1A mutant cell line (SNU449). I find this to be a weak figure, as it essentially only shows that it is possible to kill SNU449 cells with siRNAs against TEAD1 or the TEAD1 inhibitor verteporfin. It is not clear that this represents a hyper-vulnerability due to the ARID1A mutant status of these cells. For example, Wei et al. show that verteporfin reduces cell viability of SW1990 cells to a similar if not greater extent (doi: 10.1111/cas.13138) despite the fact that SW1990 are not mutant for ARID1A based on the CCLE data (<https://portals.broadinstitute.org/ccle/>). Similarly, Giraud et al. report a ~50% reduction in cell growth with 1uM verteporfin 24h (DOI: 10.1002/ijc.32667). While I was unable to obtain information on the ARID1A status in these cells, it is biased towards essentiality in the DepMap (<https://depmap.org/>) data (CERES -0.244; for comparison CERES -0.142 in SNU445 cells), suggesting that ARID1A is also functional in these cells. In their rebuttal the authors state that the time- and dose-dependence indicates specificity, which I do not find a valid argument and which to my mind does not substantiate the claim that TEAD1 inhibition is selectively toxic in the context of mutant ARID1A. This figure would greatly benefit from a very simple experiment, in which the authors reconstitute SNU449 cells with a wild-type ARID1A cDNA and show that the vulnerability to siTEAD and verteporfin is reduced.
2. New figure 5 and fig. S3 address the interactions between ARID1A and TEAD1 in Huh-7 and HLE cells. While the data seem to be clear (albeit with modest in vitro effects) for Huh-7 cells, the HLE data do not convince. In these cells, siTEAD1 and siARID1A both seem to be inhibitory for cell proliferation on their own and the effect of combined silencing of TEAD1 and ARID1A appears to additive at best (in fact, less than additive at 24h), which is not what is to be expected from a true SL interaction (defined as more than the added effect of the individual fitness defects). I had raised this additive vs. SL effect as a general concern in my previous comments and had hoped that the authors would carefully evaluate this, given that it is central to the main message of the manuscript and well within their expertise. In their rebuttal, Srivatsa et al. argue that SL requires cell death rather than inhibition of proliferation and refer to the apoptosis assay (fig. 4g; fig. 5e). Why has this assay not been performed for HLE cells?
3. In figure S3c siARID1A reduces cell proliferation (no statistical test has been performed but based on the other comparisons, it is expected to be highly significant at 24h and 48h). Using the same reagents and cell line, in figure S3d siARID1A suddenly increases cell proliferations at these time points. What does this mean? I had raised similar concerns about inconsistencies in the data in my previous review. In their rebuttal, the authors referred to batch effects, e.g. cell passages or clones). I do not find this appropriate. If the fitness effects are so variable between experiments, the general validity of these validation experiments has to be called into question.
4. I had asked about statistics and satisfyingly the authors have added a new section on this topic to their revised manuscript. In this section, it is stated that unless specified otherwise, experiments were performed in duplicates. To my knowledge, it is not appropriate to derive statistics from duplicate measurements.

Minor points:

1. Legend in figure S4i should read siTP53 instead of siURI1.

RESPONSE TO REVIEWERS' COMMENTS

We thank the reviewers for their valuable comments. We performed additional analyses and amended the manuscript extensively. Please find our point-by-point responses below.

Reviewer #1 (Remarks to the Author):

General comments

We carefully read the author's response to our initial review regarding their paper titled "Discovery of synthetic lethal interactions from large-scale pan-cancer perturbation screens". While we appreciate their efforts to address our comments, we still have some major concerns. In general, the "response to reviewer document" needs to be significantly improved. Specifically, the details regarding methods and results in a subset of responses are missing or wrongly referenced hindering our ability to judge the quality of the response. A comprehensive and well-structured response may help to overcome this. Based on this response, a major part of what was done was not clear to us. The manuscript is also not well-written.

A few main concerns regarding validations are either not satisfactorily addressed or our ability to judge is hindered due to the unclear presentation of the results. Though we acknowledge and commend the authors on their thorough effort addressing these comments partly and improve the overall manuscript. We recommend the authors to provide a revised version of the manuscript addressing our comments below before further consideration.

Specific comments

1) The simulated studies in point 2 (response to reviewer document which is in response to our 'major comments 1 and 2'), complement and supports the methodology quite well. Though we find the description of this section quite poor in both the response document as well as in the manuscript. It was hard to follow and understand what the authors did. Specifically:

- We would suggest the authors provide a high-level description of the method (please follow this in general for other analysis as well) in the results section to guide and help the reader understand.
- The motivation behind this analysis was not clear.
- How the control test was decided and What do the authors mean by saying they used the standard Wilcoxon test initially?

We thank the reviewer for the constructive comment.

- In the revised version of the manuscript, we have included the motivation and a high-level description of the method in the results under section “*SLIdR outperforms conventional tests for SL prediction*” as follows:

“To demonstrate the advantage of relative ranking over raw viability scores and to benchmark the performance of SLIdR, we compared SLIdR with existing SL prediction methods in a simulation study.”

“In addition, we included the t-test in our simulation study as we found it to be statistically more powerful than the Wilcoxon rank sum test. The simulation study focused on liver, ovarian, and bone cancers. For each cancer type, the ground truth comprised 30 random pairwise interactions between driver genes and perturbed genes. For the simulation study, we re-used the binarized mutation matrices of the corresponding cancer types and simulated the viabilities by sampling from normal distributions. In particular, for the ground truth SL pairs, the viability distribution parameters were different between mutated and WT cell lines (see Methods). Subsequently, we compared SLIdR to the Wilcoxon rank sum test and t-test on the simulated data. SLIdR significantly outperformed these tests for predicting SL interactions from simulated data in liver, ovary, and bone cancers (Supplementary Fig. S3).”

- We have tried to adopt this format of motivation and high-level description for all the other sections in the results.
- Regarding the choice of the control test, ISLE and DAISY use the Wilcoxon rank-sum test to test for conditional essentiality to define a pool of candidate SL pairs and then refine these hits using the omics data from TCGA and other sources. Therefore, we compared SLIdR to the standard Wilcoxon rank sum test. We had specified this in our earlier submission as well under the results section “*SLIdR outperforms conventional tests for SL prediction*” as follows:

“DAISY and ISLE are two popular methods that both use the Wilcoxon rank sum test for predicting candidate SL interactions from perturbation screens^{29,31}. While both of these methods are multi-step and use other data sources to predict the final pairs, their reliance on raw viability scores rather than relative viabilities from shRNA screens limits the full utilization of such screens.”

The second portion of this response is unsatisfactorily and lacks the needed depth. This step is critical to comprehensively demonstrate a validation of the methodology. Specifically, we provided a comprehensive list of gold-standard sets for the authors consideration, but they ignored mostly it. We would like to note that Horlbeck et al., 2018, a screen among the listed

ones we provided ([https://www.cell.com/cell/fulltext/S0092-8674\(18\)30735-9](https://www.cell.com/cell/fulltext/S0092-8674(18)30735-9)) contains a total of 220K pairs experimentally tested and could provide quite a large and significant set to test their method. They could have been used to generate positive and negative SL sets (gold-standard) and then used for validation, using prediction measures like ROC-AUC or Area under precision recall or some other performance measure. We must note that they are not specifically focused on driver genes and thus need to be carefully done.

From the previous list, several datasets comprised SL pairs derived from double knockdown screens performed on one or two specific cell lines. Therefore, we found that these pairs were not generalizable or robust enough to benchmark or compare. In this regard, initially, we assessed the performance of SLIdR compared to the ISLE results on a robust set of SL pairs derived from 17 different *in vitro* perturbation screens (Lee *et al.*, 2018), which included a few of the links suggested by Reviewer #1.

However, we do agree with the reviewer's comments on the issues with siRNA screens. Therefore, in this revision, we additionally compared it to a consolidated list of experimentally identified SL interactions from the CRISPR screens suggested by the reviewer (please see detailed response below). However, it should be noted that benchmarking using the CRISPR screens has also some limitations. First, the experimentally identified pairs are derived from CRISPR screens performed on very few ($n \leq 6$) highly specific cell lines, while we used a diverse set of ~373 cell lines in our cancer type-specific and pan-cancer analyses. Second, only 13 (~0.05%) common pairs between these CRISPR screens indicate that the screens are very diverse. Comparing to diverse datasets implies reduced sensitivities. Finally, a large number of screens focus mainly on identifying paralogs. Despite the lack of a consistent gold-standard set, we provide performance measures, namely, sensitivity, specificity, and accuracy in Supplementary table S4.

In addition, we performed a detailed simulation study to evaluate SLIdR's performance using ROC and PR curves and quantitatively demonstrated the advantage of SLIdR.

While we appreciate the analysis done on the gold-standard SL interactions identified by Lee *et al.*, 2018; there are some major caveats in choosing this set.

(i) Supp. Table S4 suggests that ISLE provides superior hypergeometric test results compared to SLIdR ($P = 1.44E-28$ for ISLE vs $2.76E-10$ for SLIdR). Is this an error? Also, it is surprising that ISLE identifies over 16 million SL pairs, when the original publication identifies only a few thousand? Please provide more details on how the ISLE was run?

There is no error, but since ISLE used a larger universe of potential pairs, one cannot compare the two p-values. Both the SLIdR and ISLE pairs are enriched in the

gold-standard SL interactions identified by Lee *et al.* Regardless, due to the differences mentioned by the reviewer and the following reasons, we decided not to compare to ISLE anymore.

- The results from our previous revision seem to be confusing and misleading the reader. ISLE uses Wilcoxon rank sum test in the first step to define an initial pool and then refines these hits using other omics data, resulting in fewer final SL pairs. We did not run ISLE and instead used the statistics from their paper from the first step (Initial pool set I as described in their paper). Originally, we compared SLIdR only to the candidate pairs from the first step of ISLE as it uses the standard Wilcoxon rank sum test in the first step. Since the purpose of the comparison was to show that using SLIdR on ranked viabilities works better than using the standard Wilcoxon rank sum test, we compared it to the results from this step. In the revised version of the manuscript, we used the simulation studies to benchmark and illustrated the advantage of using SLIdR on ranked viabilities.

Furthermore, unlike ISLE, our method is not integrative and does not use other omics data. In the previous version, we had emphasized this in our discussion and suggested using SLIdR in tandem with methods like ISLE in the future. In the discussion, we had written,

“While SLIdR was successful in identifying SL pairs, there are several limitations: (1) In its current scope, SLIdR primarily focuses on identifying SL partners from perturbation screening data, copy number, and mutation data. Consequently, it failed to recover the predictions based on expression and pathway reported by McDonald *et al.*²⁷, emphasizing the importance of integrating other omics data. Thus, extending SLIdR to incorporate multi-omics and pathway data or using it in tandem with methods like ISLE could further improve its overall performance.”

- Since we could not find the entire list of candidate pairs for ISLE, we could not quantitatively assess the performance of ISLE against the consolidated list of experimentally identified SL pairs from CRISPR screens.

(ii) The initial set chosen comprises siRNA screens performed in the early decade. The quality of these screens were shown to be very poor due to the off-target effect of siRNA and thus not appropriate for gold-standard set (PMCID: 10.1186/s11658-019-0196-3 , <https://doi.org/10.1038/s41598-017-18551-z> , <https://doi.org/10.1371/journal.pbio.2003213>). Considering this, we think Lee *et al.*'s

step to consider this SL set as an initial pool is okay, though choosing this set as gold-standard is not justified.

Thank you for this comment. We also evaluated the performance of SLIdR on 10 combinatorial CRISPR screens. As we mentioned earlier, benchmarking on CRISPR screens has its own limitations. Hence, we decided to benchmark our method on (i) 17 siRNA screens reported by Lee *et al.* and (ii) 10 combinatorial CRISPR screens. We have updated the "SLIdR recovers experimentally identified SL interactions" section accordingly.

We suggest the authors to please run a validation step considering some of the following screens. Please find below a list of recent combinatorial CRISPR screening studies:

- Gonatopoulos-Pournatzis, T., Aregger, M., Brown, K.R. et al. Genetic interaction mapping and exon-resolution functional genomics with a hybrid Cas9–Cas12a platform. *Nat Biotechnol* 38, 638–648 (2020)
- Aregger, M., Lawson, K.A., Billmann, M. et al. Systematic mapping of genetic interactions for de novo fatty acid synthesis identifies C12orf49 as a regulator of lipid metabolism. *Nat Metab* 2, 499–513 (2020).
- Thompson, N.A., Ranzani, M., van der Weyden, L. et al. Combinatorial CRISPR screen identifies fitness effects of gene paralogues. *Nat Commun* 12, 1302 (2021).
- DeWeirdt, P.C., Sanson, K.R., Sangree, A.K. et al. Optimization of AsCas12a for combinatorial genetic screens in human cells. *Nat Biotechnol* 39, 94–104 (2021).
- Dede, M., McLaughlin, M., Kim, E. et al. Multiplex enCas12a screens detect functional buffering among paralogs otherwise masked in monogenic Cas9 knockout screens. *Genome Biol* 21, 262 (2020).
- Gier, R.A., Budinich, K.A., Evitt, N.H. et al. High-performance CRISPR-Cas12a genome editing for combinatorial genetic screening. *Nat Commun* 11, 3455 (2020).
- Replogle, J.M., Norman, T.M., Xu, A. et al. Combinatorial single-cell CRISPR screens by direct guide RNA capture and targeted sequencing. *Nat Biotechnol* 38, 954–961 (2020).
- DeWeirdt, P.C., Sangree, A.K., Hanna, R.E. et al. Genetic screens in isogenic mammalian cell lines without single cell cloning. *Nat Commun* 11, 752 (2020).

- Liu, J. et al. Pooled library screening with multiplexed Cpf1 library. *Nat Commun* 10, 3144 (2019).
- Zhao, Y., Tyrishkin, K., Sjaarda, C. et al. A one-step tRNA-CRISPR system for genome-wide genetic interaction mapping in mammalian cells. *Sci Rep* 9, 14499 (2019).
- Norman, Horlbeck, Replodge, Ge, Xu, Jost, Gilbert, Weissman, Exploring genetic interaction manifolds constructed from rich single-cell phenotypes, *Science* 786-793 (2019).
- Boettcher, M. et al. Dual gene activation and knockout screen reveals directional dependencies in genetic networks. *Nat. Biotechnol.*36, 170–178 (2018).
- Zhao, Badur, Luebeck, Magaña, Birmingham, Sasik, Ahn, Ideker, Metallo, Mali, Combinatorial CRISPR-Cas9 Metabolic Screens Reveal Critical Redox Control Points Dependent on the KEAP1-NRF2 Regulatory Axis, *Molecular Cell*, Volume 69, Issue 4, 699-708.e7 (2018).
- Horlbeck, M. A. et al. Mapping the genetic landscape of human cells. *Cell*174, 953–967.e22 (2018).
- Han, K. et al. Synergistic drug combinations for cancer identified in a CRISPR screen for pairwise genetic interactions. *Nat. Biotechnol.*35, 463–474 (2017).
- Najm, F. J. et al. Orthologous CRISPR–Cas9 enzymes for combinatorial genetic screens. *Nat. Biotechnol.*36, 179–189 (2017).
- Shen, J. P. et al. Combinatorial CRISPR–Cas9 screens for de novo mapping of genetic interactions. *Nat. Methods*14, 573–576 (2017).
- Wong, A. S. L. et al. Multiplexed barcoded CRISPR-Cas9 screening enabled by CombiGEM. *Proc. Natl Acad. Sci. USA*113, 2544–2549 (2016).

We thank the reviewer for the constructive comment and the extensive list. In the revised version of the manuscript, we included 10 of the recommended CRISPR screens for validation. We updated the results and corresponding methods in the new version of the manuscript.

We updated the methods under section “*Comparison to experimentally identified SL interactions*” as follows:

“To compare SLiDRs predictions with established SL interactions, we focused on (i) 6,033 experimentally identified SL interactions from 17 siRNA screens reported by Lee et al.³¹, and (ii) 24,651 experimentally identified SL interactions from 10 combinatorial CRISPR screens^{17–26}

(details in Supplementary notes). Since we applied SLIdR to the Project DRIVE dataset, we excluded the experimentally identified SL interactions that were not in the set of possible pairwise interactions in the DRIVE dataset (9,443,304), yielding 978 and 1,301 unique experimentally identified SL interactions in the siRNA and CRISPR screens, respectively. The retained 1,301 unique experimentally identified SL interactions included pairs from 8 of the 10 CRISPR screens and four shared SL interactions (Supplementary Table S4). Finally, we re-ran SLIdR in the pan-cancer setting by relaxing the significance level to 5% and used a hypergeometric test to assess whether the overlap between predicted and experimentally identified SL pairs was significant. We also computed the sensitivity, specificity, and accuracy in both sets.”

We provided additional information on the methods in supplementary notes under section “*Consolidating experimentally identified SL interactions from CRISPR screens*” as follows:

“To compare SLIdRs predictions with established SL interactions, we focused on experimentally identified SL interactions from siRNA screens and combinatorial CRISPR screens. While the list of interactions for the former was obtained from Lee et al.’s³¹ study, for CRISPR based SL interactions, we consolidated 24,651 experimentally identified SL interactions from 10 combinatorial CRISPR screens^{17–26} with 13 shared interactions across the screens. Since, SLIdR predicts only SL interactions, we retained pairs with negative genetic interactions (GI) scores from the works of Horlbeck et al.¹⁹ and Norman et al.²². DeWeirdt et al.²⁶ provided a ranked list of SL partners for *PARP1*, *BCL2L1*, and *MCL1* and we retained the first 100 for each. It should be noted that majority of these experiments were performed in a selective few cell lines and several studies focused only on identifying paralog pairs, and SL partners for specific driver genes.”

We updated the corresponding results under section “*SLIdR recovers experimentally identified SL interactions*” as follows:

“Going beyond simulated data, we also evaluated the overlap of SLIdRs pan-cancer predictions with experimentally identified SL interactions from (i) 17 siRNA screens reported by Lee *et al.*³¹, and (ii) 10 combinatorial CRISPR screens^{17–26}. Excluding the experimentally identified SL interactions that were not in the set of possible pairwise interactions in the DRIVE dataset, we focused on 978 and 1,301 unique experimentally identified SL interactions in the siRNA and CRISPR screens, respectively, for these comparisons (**see Methods**). The retained 1,301 unique experimentally identified SL interactions included pairs from 8 of the 10 CRISPR screens. Only four interactions were shared across these 8 combinatorial CRISPR screens. In contrast, SLIdR recovered a significant fraction of established SL interactions (hypergeometric p-values $< 10^{-9}$)

with sensitivities of 12.3 % and 11.45% in the siRNA and CRISPR screens, respectively. With accuracies of ~93% across both screens, these results comprehensively validated SLIdR predictions (**Supplementary Table S4**)."

2) Regarding the authors' response to our original "major comment 4": The drug response prediction results are poorly presented hindering our ability to judge the results and are not convincing. Specifically

- Multiple hypothesis correction (FDR) needs to be done, otherwise these p-values could be false positives.

We thank the reviewer for the constructive comment. In the revised version of the manuscript, we have included FDR corrected p-values (q-values) in Supplementary Table S3. However, given the heterogeneity and noise of the screens, we do have fewer hits after FDR correction and miss some well-known SL pairs such as *KEAP1-NFE2L2* in the pan-cancer analysis. Therefore, we decided to discuss both the uncorrected and FDR corrected hits in the results with an explicit mention for each case.

- Appropriate control experiments using random SL pairs could also be carried out.

Thank you for this suggestion. In the revised version of the manuscript, we have performed control experiments through permutation tests on 1000 sets of random gene pairs. We have added the results and corresponding methods in the new version of the manuscript.

We updated the methods under section "*Control experiments on PRISM screen*" as follows:

"To assess the robustness of the PRISM validation results of SLIdR hits, we compared them to the PRISM results obtained from 1000 sets of random SL pairs. The same process was followed for both pan-cancer and cancer type-specific settings. We first retained only those perturbed genes with at least one matching drug compound in the PRISM screen for each cancer. We then generated a set of all possible pairwise interactions (U) between all the driver genes and druggable perturbed genes and filtered out all the pairs exhibiting oncogene-addiction. Subsequently, we ran permutation tests on random sets of gene pairs. For each run, we sampled K SL pairs from U at random, where K is the number of SL pairs predicted by SLIdR and tested on PRISM screens. For each sampled SL pair, we stratified the cell lines based on the mutation status of the driver gene into WT and mutated cell lines. Then, we tested whether the drug compound targeting the corresponding SL partner gene reduced the

viabilities of mutated cell lines compared to WT cell lines using a one-sided t-test. Since multiple drug compounds could target the same SL partner gene, we tested the differential drug response for each drug compound for a given SL pair. We repeated this for all SL pairs and counted the SL pairs with a significant difference in drug response (significance level of $\alpha = 0.1$). Finally, we computed the empirical p-values by comparing the number of significant drug responses of SLIdR hits and those of random sets.”

We updated the corresponding results under section “*Enrichment of pan-cancer SL interactions by SLIdR*”

“As a control experiment, we performed permutation tests across 1000 sets of random gene pairs and found these validated hits to be significant (empirical p-value = 0.005), thereby confirming the predictions from SLIdR (see Methods).”

And also under section “*Enrichment of cancer type-specific SL interactions by SLIdR*”.

“We also performed control experiments based on permutation tests of 1000 sets of random gene pairs and found some evidence for these hits in pancreatic, skin, stomach, and renal cancers (empirical p-value = {0.024, 0.076, 0.002, 0.075}, respectively), which reached statistical significance after multiple testing correction for stomach cancer (q-value = 0.02). These results demonstrate that SLIdR is capable of finding known and novel cancer type-specific SL pairs.”

- Fig. 2d is referenced in the results of pan-cancer testing. We did not find any pan-cancer results in figure 2d. Please provide this separately.

We would like to clarify that hits in figure 2d are indeed from the pan-cancer analysis. Each panel of the figure represents all the pan-cancer SL pairs of a specific driver gene, where the driver gene is shown at the x-axis top, and the corresponding SL partner genes are shown at the x-axis bottom. For example, the fourth panel indicates that the driver gene *BRAF* has five SL partners, namely *CYP2B6*, *ESPN*, *HTR2C*, *PAK1*, and *PMP22*. The primary purpose of this figure is to illustrate the sensitivities of identified pan-cancer SL pairs in cell lines grouped by primary sites (rows). To clarify this, we modified the caption of the figure. Additionally, the corresponding part of the last paragraph of the “*Enrichment of pan-cancer SL interactions by SLIdR*” section now reads:

“To assess the differential sensitivities of the predicted pan-cancer hits based on primary sites, we computed the SLIdR p-values for these predicted pan-cancer SL pairs in subsets of cell lines grouped by primary sites. We found that a sizable fraction of the pan-cancer signals are indeed cancer type-specific (Fig. 2d).”

- In addition, Fig. 2d needs improvement. Specifically, most of the p-values are blue colored and based on the current legend it is not clear whether they are significant or not. Fig. 3b does not indicate drug response validation. We did not find Fig. 3b to be informative, a simpler figure showing the number of drugs which pass the FDR threshold can be shown for various drugs or volcano plots showing controls and SLs together providing significance and effect size together. Drug response prediction can be shown either in terms of p-values (after FDR correction), or ROC-AUCs; based on whatever the authors deem fit.

We thank the reviewer for this comment.

In the revised version of the manuscript, we have made the legend more granular in Figure 2d. As mentioned in the previous comment, the goal of this figure is to illustrate the sensitivities of identified pan-cancer SL pairs in cell lines grouped by primary sites. The blue tiles in the heatmap suggest that the synthetic lethal signal is not specific to a primary site. In contrast, the green/yellow tiles indicate that the majority of the signal is specific to a primary site. For example, *BRAF*-associated SL interactions are mostly specific to skin cancer cell lines.

In the revised version of the manuscript, we have retained Figure 3b as it summarizes the cancer type-specific hits with literature evidence and evidence in PRISM screens before correction. This has been explicitly mentioned in the caption. We also acknowledge the reviewer’s suggestion and have included Supplementary Fig. S2 showing the number of significant candidate drugs ($\alpha = 0.1$) in pan-cancer and cancer type-specific settings on the primary PRISM repurposing dataset. This figure also highlights the number of drugs that are significant after FDR correction (q-values ≤ 0.2). Additionally, drug response prediction results with p-values and q-values have been summarized in Supplementary Table S3.

In general, we understand that drug response is a hard question and therefore this is not critical. Thus, it is okay even if SLIdR does not work well in predicting drug response, given the heterogeneity and noise of screens or provides a very low coverage after FDR is correct, it needs to be explicitly mentioned.

Thank you for this comment. It is true that given the heterogeneity and noise of the screens, we do have fewer hits after FDR correction. Therefore, in the revised version of the manuscript, we discussed both the uncorrected and FDR corrected hits in the results with an explicit mention for each case.

3) We are fine with the rest of the responses.

We thank and commend the authors for their effort in addressing these comments and improving their manuscript.

Reviewer #3 (Remarks to the Author):

In their revised manuscript, Srivatsa et al. have addressed several of my earlier concerns, which in my eyes has strengthened their work. Most importantly, they have elaborated on what distinguishes their method from other methods of SL pair prediction and have included additional validation. They have addressed my concern about AXIN1/URI1 by moving these data to the supplement and introduced a new SL pair ARID1A/TEAD1 for the validation experiments and in their rebuttal state that investigation of the 'high profile' SL pairs identified by their approach would be beyond the scope of the study. Overall, the authors are to be complemented for their additional efforts and the new data help to make a stronger case for their method. However, the introduction of these new data also comes with some new concerns that should be addressed prior to publication.

Major points:

1. The new figure 4 is entirely dedicated to studying the effect of TEAD1 inhibition in an ARID1A mutant cell line (SNU449). I find this to be a weak figure, as it essentially only shows that it is possible to kill SNU449 cells with siRNAs against TEAD1 or the TEAD1 inhibitor verteporfin. It is not clear that this represents a hyper-vulnerability due to the ARID1A mutant status of these cells. For example, Wei et al. show that verteporfin reduces cell viability of SW1990 cells to a similar if not greater extent (doi: 10.1111/cas.13138) despite the fact that SW1990 are not mutant for ARID1A based on the CCLE data (<https://portals.broadinstitute.org/ccle/>). Similarly, Giraud et al. report a ~50% reduction in cell growth with 1uM verteporfin 24h (DOI: 10.1002/ijc.32667). While I was unable to obtain information on the ARID1A status in these cells, it is biased towards essentiality in the DepMap (<https://depmap.org/>) data (CERES -0.244; for comparison CERES -0.142 in SNU445 cells), suggesting that ARID1A is also functional in these cells. In their rebuttal the authors state that the time- and dose-dependence indicates specificity, which I do not find a valid argument and which to my mind does not substantiate the

claim that TEAD1 inhibition is selectively toxic in the context of mutant ARID1A. This figure would greatly benefit from a very simple experiment, in which the authors reconstitute SNU449 cells with a wild-type ARID1A cDNA and show that the vulnerability to siTEAD and verteporfin is reduced.

We thank the reviewer for the constructive comment. In the revised version of the manuscript, we have performed the experiment suggested by the reviewer and shown that reconstitution of wild-type ARID1A via overexpression of wild-type ARID1A cDNA desensitized SNU449 response to TEAD1 inhibition via treatment with verteporfin. Our results support our hypothesis of a synthetic lethal interaction between mutant ARID1A and TEAD1 in liver cells.

We updated the methods under section "*Transient ARID1A overexpression*" as follows:

"For ARID1A overexpression, pcDNA6-ARID1A (#39311)⁶⁷ was obtained from Addgene (<https://www.addgene.org>), and the empty control vector was pCDNA6-V5/His.b (ThermoFisher Scientific). The expression vectors were transiently transfected using the Lipofectamine 3000 transfection reagent (ThermoFisher Scientific) following the manufacturer's instructions."

Subsequently, we updated the corresponding results under section "*SLiDR identified two novel targets in hepatocellular carcinoma*" as follows:

"Furthermore, to demonstrate that indeed SNU449 sensitivity to verteporfin was specifically dependent on the presence of mutant ARID1A, we rescued ARID1A wild-type expression in SNU449 cells. Indeed, we observed that rescuing wild-type ARID1A desensitized cells to verteporfin thus indicating that the presence of a mutant ARID1A in SNU449 is essential to confer sensitivity towards TEAD1 inhibition (**Fig. 4h**)."

2. New figure 5 and fig. S3 address the interactions between ARID1A and TEAD1 in Huh-7 and HLE cells. While the data seem to be clear (albeit with modest in vitro effects) for Huh-7 cells, the HLE data do not convince. In these cells, siTEAD1 and siARID1A both seem to be inhibitory for cell proliferation on their own and the effect of combined silencing of TEAD1 and ARID1A appears to additive at best (in fact, less than additive at 24h), which is not what is to be expected from a true SL interaction (defined as more than the added effect of the individual fitness defects). I had raised this additive vs. SL effect as a general concern in my previous comments and had hoped that the authors would carefully evaluate this, given that it is central to the main message of the manuscript and well within their expertise. In their rebuttal, Srivatsa et al. argue that SL requires cell death rather than inhibition of proliferation and refer to the apoptosis assay (fig. 4g; fig. 5e). Why has this assay not been performed for HLE cells?

We thank the reviewer for the comment. While the putative additive effect between *ARID1A* and *TEAD1* shown in HLE is actually transient, with no significant loss in cell viability later than 48h in either *ARID1* or *TEAD1* silenced alone, we agree with the reviewer's concerns and, as requested by the reviewer, we have performed the apoptosis assay in HLE cells (**Supplementary Fig. S4f**). Similar to the results obtained in Huh7 cells, HLE cells for which *ARID1A* expression was silenced had a significant decrease in cell viability upon treatment with verteporfin compared to control cells transfected with siRNA control. No difference in live or apoptotic cells was detected between control or *ARID1A* silenced cells. This experiment indicates that *ARID1A* silencing did not affect cell viability *per se* but only in combination with *TEAD1* inhibition, thus excluding the hypothesis of an additive effect and rather indicating that loss of *ARID1A* sensitized cells to *TEAD1* inhibition, supporting the synthetic lethal interaction between the two genes.

3. In figure S3c siARID1A reduces cell proliferation (no statistical test has been performed but based on the other comparisons, it is expected to be highly significant at 24h and 48h). Using the same reagents and cell line, in figure S3d siARID1A suddenly increases cell proliferations at these time points. What does this mean? I had raised similar concerns about inconsistencies in the data in my previous review. In their rebuttal, the authors referred to batch effects, e.g. cell passages or clones). I do not find this appropriate.

We thank the reviewer for the comment. We have repeated the experiments adding more technical and biological replicates. We combined the previous results and the newly obtained ones. As shown in the new **Supplementary Fig. S4c**, the sole inhibition of *TEAD1* or *ARID1A* does not significantly reduce cell proliferation. Importantly, similarly to **Supplementary Fig. S4d**, we observed an increase in cell proliferation upon *ARID1A* silencing.

4. I had asked about statistics and satisfyingly the authors have added a new section on this topic to their revised manuscript. In this section, it is stated that unless specified otherwise, experiments were performed in duplicates. To my knowledge, it is not appropriate to derive statistics from duplicate measurements.

We thank the reviewer for pointing this out. In the revised version of the manuscript, the experiments were performed in triplicate or more for consistency. A data source table is submitted with all the raw data for clarity.

Minor points:

1. Legend in figure S4i should read siTP53 instead of siURI1.

Thank you for pointing this out! This typo has been corrected (now **Fig. S5i**).

REVIEWERS' COMMENTS

Reviewer #1 (Remarks to the Author):

The authors have provided a well-structured and clear response. Most importantly, they have now thoroughly validated their method on more than ten screens. We would recommend the authors, please make sure that both data and code are accessible to the reader so that they can utilize this information. We believe this could be, in addition to the method, very useful to community.

Reviewer #3 (Remarks to the Author):

In their revised manuscript, Srivatsa et al. have performed some additional experiments to address the concerns I had raised in previous review rounds. The authors are to be commended for their additional efforts which have improved the manuscript. However, a few issues remain.

The requested ARID1A 'addback' experiment has been performed, which I appreciate, and the results are shown in Fig. 4h. All in all, there seems to be a trend but the effects are modest and errors are unfortunately large. In an ordinary 2-way ANOVA (which I would consider a more appropriate test here) a significant result is only gained for the 1 μ M condition ($P = 0.047$).

I had previously pointed out inconsistencies in the results obtained in different experiments. Srivatsa et al. have added more replicates in the revised manuscript, presumably also to address my previous point that it is not appropriate to derive statistics from only duplicate measurements. In contrast to earlier versions of the manuscript, siTEAD1 now no longer reduces cell proliferation in S3c (now S4c) and is now consistent with panel d. While this reduces the previous inconsistency, no explanation was offered as to why the results changed between submissions. Given that the differential fitness effects are the core of the paper, an explanation would go a long way to inspire confidence in the presented data.

Along similar lines, I applaud the authors for now including source data of the experiments. However, in the source data file, it appears as if in all experiments individual data points seem to have been omitted (gaps). Surely the experiments were not designed this way? Generally, data exclusion is very problematic unless there is a clear justification for it. In case individual replicates were omitted, I could not find corresponding explanations in the 'data exclusion' section of the Reporting Summary.

RESPONSE TO REVIEWERS' COMMENTS

We thank the reviewers for their valuable comments. Please find our point-by-point responses below.

REVIEWERS' COMMENTS

Reviewer #1 (Remarks to the Author):

The authors have provided a well-structured and clear response. Most importantly, they have now thoroughly validated their method on more than ten screens. We would recommend the authors, please make sure that both data and code are accessible to the reader so that they can utilize this information. We believe this could be, in addition to the method, very useful to community.

Thank you for your favorable feedback. We have included the data sources and code in the Data availability and Code availability sections, respectively. The code has been and will continue to be in a public Github repository.

Reviewer #3 (Remarks to the Author):

In their revised manuscript, Srivatsa et al. have performed some additional experiments to address the concerns I had raised in previous review rounds. The authors are to be commended for their additional efforts which have improved the manuscript. However, a few issues remain.

We thank the reviewer for your favorable feedback and for appreciating our work.

The requested ARID1A 'addback' experiment has been performed, which I appreciate, and the results are shown in Fig. 4h. All in all, there seems to be a trend but the effects are modest and errors are unfortunately large. In an ordinary 2-way ANOVA (which I would consider a more appropriate test here) a significant result is only gained for the 1uM condition ($P = 0.047$).

We thank the reviewer for the comment. The two-way ANOVA can assess the main effects of the two variables, namely the Verteporfin dosage and the experiment (CRTox and ARID1Aox), and their interaction on cell viability. As a subsequent analysis, one can compare means within each dosage. However, our goal in this figure was to compare cell viabilities between CRTox and ARID1Aox for each dosage of Verteporfin. We, therefore, think the t-test is a suitable choice here and prefer to keep the existing figure and results as it is.

I had previously pointed out inconsistencies in the results obtained in different experiments. Srivatsa et al. have added more replicates in the revised manuscript, presumably also to address my previous point that it is not appropriate to derive statistics from only duplicate measurements. In contrast to earlier versions of the manuscript, siTEAD1 now no longer reduces cell proliferation in S3c (now S4c) and is now consistent with panel d. While this reduces the previous inconsistency, no explanation was offered as to why the results changed between submissions. Given that the differential fitness effects are the core of the paper, an explanation would go a long way to inspire confidence in the presented data.

We thank the reviewer for the comment and we apologize for not explaining further in the previous rebuttal letter. As the reviewer pointed out, we added replicates to increase the robustness of our results. The results from the new experiments were in line with our main results demonstrating the synthetic lethal interaction of *ARID1A* and *TEAD1*. Indeed, in the earlier version, siTEAD1 reduced cell proliferation and this is no longer the case. We believe that this observation was due to the toxicity induced by the transfection of siRNA. Our new results (now Supplementary Figure 4f) also show that the transfection of siARID1A did not impact cell viability.

Along similar lines, I applaud the authors for now including source data of the experiments. However, in the source data file, it appears as if in all experiments individual data points seem to have been omitted (gaps). Surely the experiments were not designed this way? Generally, data exclusion is very problematic unless there is a clear justification for it. In case individual replicates were omitted, I could not find corresponding explanations in the 'data exclusion' section of the Reporting Summary.

We thank the reviewer for the comment and apologize for not providing the information required. Some of the data points from technical replicates were removed due to likely cell plating and drug dosage anomalies of the multichannel pipette, or were removed as they were impossible values that could result from errors in the execution of the experiments or data entry. However, we always make sure that every technical replicate contains at least 3 measurements in order to perform appropriate statistical tests. In the revised version of the Reporting Summary, we have updated the 'data exclusion' section.